# Chronic kidney disease prediction using boosting techniques based on clinical parameters

**Shahid Mohammad Ganie** **[1], Pijush Kanti Dutta Pramanik[2], Saurav Mallik[3], Zhongming Zhao[4]***

**1** AI Research Centre, School of Business, Woxsen University, Hyderabad, Telangana, India, **2** School of Computing Science and Engineering, Galgotias University, Greater Noida, Uttar Pradesh, India, **3** Department of Environmental Health, Harvard T. H. Chan School of Public Health, Boston, MA, United States of America, **4** Center for Precision Health, School of Biomedical Informatics, The University of Texas Health Science Center at Houston, Houston, TX, United States of America

* zhongming.zhao@uth.tmc.edu

**Data Availability Statement:** All relevant data can be found at https://archive.ics.uci.edu/ml/datasets/chronic_kidney_disease.

**Funding:** Zhongming Zhao was partially supported by his startup fund from The University of Texas

## Abstract

Chronic kidney disease (CKD) has become a major global health crisis, causing millions of yearly deaths. Predicting the possibility of a person being affected by the disease will allow timely diagnosis and precautionary measures leading to preventive strategies for health. Machine learning techniques have been popularly applied in various disease diagnoses and predictions. Ensemble learning approaches have become useful for predicting many complex diseases. In this paper, we utilise the boosting method, one of the popular ensemble learnings, to achieve a higher prediction accuracy for CKD. Five boosting algorithms are employed: XGBoost, CatBoost, LightGBM, AdaBoost, and gradient boosting. We experimented with the CKD data set from the UCI machine learning repository. Various preprocessing steps are employed to achieve better prediction performance, along with suitable hyperparameter tuning and feature selection. We assessed the degree of importance of each feature in the dataset leading to CKD. The performance of each model was evaluated with accuracy, precision, recall, F1-score, Area under the curve-receiving operator characteristic (AUC-ROC), and runtime. AdaBoost was found to have the overall best performance among the five algorithms, scoring the highest in almost all the performance measures. It attained 100% and 98.47% accuracy for training and testing sets. This model also exhibited better precision, recall, and AUC-ROC curve performance.

## 1. Introduction

Chronic kidney disease (CKD) has become very common across races [1], resulting in millions of deaths worldwide annually [2]. Proper diagnosis and timely treatment are major concerns in most developing countries. CKD mostly hits older people [3, 4], and by 2050, the number of people aged 65 years and above is estimated to increase to 1.5 billion from 703 million in 2019,

Health Science Center at Houston, Houston, Texas, USA.

**Competing interests:** The authors have declared that no competing interests exist.

with a more than double growth rate [5]. This will put a significant additional burden on healthcare services across the countries [6].

According to a study by the Center for Disease Control and Prevention, in 2017, approximately thirty million people in the U.S. alone were affected by CKD [7], which has been increased to 37 million in 2021 [8]. Moreover, most people are not aware of being infected by CKD. Traditionally, doctors confirm the CKD for any patient based on some clinical tests such as estimating glomerular filtration rate (GFR) from a filtration marker (e.g., serum creatinine or cystatin C) or through a urine test, detecting the presence of albumin and/or protein [9–11]. However, these tests may not always give accurate results, leading to the wrong diagnosis.

CKD can be mitigated to some extent if the possibility of it can be predicted beforehand for the suspected patients [12, 13]. This would allow healthcare professionals to deliver better services by embracing precautionary measures and early diagnosis and treatment. Machine learning algorithms have been popularly used in several disease diagnoses and predictions [14–17]. For CKD prediction also, various such techniques have been explored [18–22]. Machine learning algorithms are powerful for analysing large and complex datasets and identifying patterns and relationships that may not be apparent to human experts. In the context of CKD prediction, machine learning has the potential to improve accuracy and reduce costs by identifying early signs of disease progression and predicting the risk of developing CKD in at-risk populations.

However, traditional machine learning techniques suffer from some crucial limitations, including [23, 24]:

- Overfitting, where the algorithm becomes too specialised to the training data and fails to generalise to new data.

- Large, high-quality datasets are needed to train and validate the algorithms, which can be challenging to obtain in some clinical settings.

- Training and evaluating machine learning algorithms may require considerable computational time and resources, especially for large datasets.

- High dependency on the quality and quantity of data available for training. If the data is incomplete, biased, or otherwise of poor quality, the resulting algorithm will be inaccurate or may not work at all.

- The machine learning algorithms can inadvertently incorporate biases present in the training data, leading to unfair or discriminatory outcomes.

Recently, ensemble learning techniques have shown great promise in improving the accuracy, robustness, and generalizability of predictive models, making them valuable in many fields, including healthcare, finance, marketing, social media analytics, etc. The ensemble learning approaches are gaining attention for disease prediction with higher accuracy [25–31]. Among the ensemble learning techniques such as boosting, bagging, and stacking, boosting algorithms can reduce the training error (bias) and testing error (variance).

In this paper, we design a novel CKD prediction model using boosting algorithms. We aim to improve the performance of the disease prediction model over similar existing works. The contributions of this paper are summarised as follows.

- Exploratory data analysis is performed to transform the considered dataset for better experimental usability.

- Hyperparameter techniques, such as standardisation, normalisation, feature selection, and fine-tuning, are employed to achieve optimal results.

- The attribution of existing dataset features to disease prediction is assessed.

- Five boosting algorithms are individually applied to build the prediction model.

- The prediction performances of the five boosting algorithms are evaluated and compared.

- Our model achieved better accuracy and runtime than other machine learning-based CKD prediction models in method evaluation.

## 2. Related work

As mentioned above, machine learning has been extensively used for various disease diagnoses and predictions [17, 32, 33]. To improve the performance of these models, several machine learning techniques are combined to extract the advantages of each of them. This ensemble approach has gained acceptance and popularity after successful implementations for the prediction, detection, diagnosis, and prognosis of different diseases, such as heart disease [34, 35], breast cancer [36], skin disease [37], thyroid disease [38], myocardial infarction [39], Alzheimer's disease [40], etc. For CKD prediction, several prediction techniques and models have already been proposed [41]. In the following, we briefly review some notable experiments for the diagnosis and prediction of CKD using ensemble learning techniques.

For CKD prediction, Kumar et al. [42] proposed an ensemble learning approach that comprises a support vector machine (SVM), decision tree, C4.5 decision tree, particle swarm optimisation - multilayer perceptron (PSO-MLP), and artificial bee colony C4.5. The prediction process has two steps–i) in the first step, weak decision tree classifiers are obtained from C4.5, and ii) in the second step, the weak classifiers are combined with the weighted sum to get the final output from the classifier, attaining accuracy of 92.76%. Pal [43] developed a bagging ensemble method comprising a decision tree, SVM, and logistic regression to predict CKD. The best accuracy of 95.92% was achieved in the case of the decision tree. Hasan and Hasan [44] proposed an ensemble method for kidney disease diagnosis. They used adaptive boosting (AdaBoost), bootstrap aggregating, extra trees, gradient boosting, and random forest to build their prediction model. They performed tenfold cross-validation to validate the results. The highest accuracy of 99% was attained with adaptive boosting. For CKD detection, Wibawa et al. [45] developed an ensemble learning method that comprises three stages. In the first stage, base classifiers like Naive Bayes, SVM, and k nearest neighbour (kNN) were used. Correlation-based feature selection (CFS) was combined with the base classifiers mentioned above in the second stage. In the third stage, they used CFS with AdaBoost, achieving the highest accuracy of 98.01%. For CKD diagnosis, Jongbo et al. [1] built an ensemble learning model through bagging and random subspace based on three base classifiers–kNN, naïve Bayes, and decision tree. Data preprocessing was done to mitigate the missing value issue and data normalisation for scaling the independent variables within a certain range. The random subspace gained better performance than bagging in most performance measure metrics. It achieved an accuracy of 98.30% when combined with the decision tree method. To detect CKD, Ebiaredoh-Mienye et al. [46] combined the information-gain-based feature selection technique with the proposed cost-sensitive AdaBoost (C.S. AdaBoost), intending to save CKD screening time and cost. They trained the proposed C.S. AdaBoost with the reduced feature set, which attained a maximum accuracy of 99.8%. Emon et al. [47] used various boosting techniques to predict the risk of CKD progression among patients. The authors applied the principal component analysis (PCA) method to get the optimal feature set and attained the highest accuracy rate of 99.0% using random forest (R.F.). Ramaswamyreddy et al. [48] used wrapper methods

along with bagging and boosting models to develop a CKD prediction model, attaining an accuracy of 99.0% with gradient boosting. However, the authors did not evaluate their model using other performance measure metrics.

## 3. Research methodology

This section briefly discusses the research steps followed and the ensemble learning techniques used in the experiment.

### 3.1 Research workflow

The workflow of the proposed work is shown in Fig 1. We performed exploratory data analysis on the considered dataset for better quality assessment. In this phase, missing values are identified and replaced using data imputation methods. The interquartile range (IQR) method is used to detect outliers present in the dataset. Some other required libraries are executed to check the corrupt data, if any, in the dataset. Also, standardisation, normalisation, feature selection, and tuning are made during the prediction model development process using five boosting algorithms. The dataset was split into training (60%) and test (40%) subsets. The results are assessed through various performance metrics.

### 3.2 Boosting algorithms

Ensemble learning is a method that combines different traditional machine learning approaches to enhance the performance of the prediction model [49]. Various ensemble learning approaches are proposed [50, 51]. Boosting algorithm is one of the effective approaches in the ensemble learning family. In the literature, several boosting algorithms can be found [52, 53]. In this experiment, specifically for CKD prediction, we considered the following five ensemble learning based boosting algorithms:

**XGBoost.** XGBoost (eXtreme gradient boosting) works by combining different kinds of decision trees (weak learners) to calculate the similarity scores independently [54]. It helps to overcome the problem of overfitting during the training phase by adapting the gradient descent and regularisation process. The mathematical formula for the XGBoost algorithm is shown in Eq 1.

$$f_\theta(x) = \sum_{m=1}^{T} \gamma_m h_m(x; \theta_m) = \sum_{m=1}^{T} \gamma_m l\left(x \in R_{jm}\right) \tag{1}$$

where $f_\theta(x)$ is XGBoost model with parameters $\theta$, $h_m$ is the $m^{th}$ weak decision tree with parameters $\theta_m$, and $\gamma_m$ is the weight associated with $m^{th}$ tree. $T$ denotes the number of decision trees, $l$ denotes the loss function, and $R_{jm}$ is an indicator function that returns 1 if $x$ is in region $R_{jm}$, otherwise 0.

**CatBoost.** CatBoost (categorical boosting) is faster than other boosting algorithms as it does not require the exploration of data preprocessing [55]. It is used to deal with high cardinality categorical variables. For low cardinality variables, one-hot encoding techniques are used for conversion. The objective function for the CatBoost algorithm is defined using Eq 2.

$$L(y, f(x)) = \sum_{i=1}^{N} l\left(y_i, f(x_i) + \frac{\lambda}{2} \sum_{j=1}^{P} w_j^2\right) \tag{2}$$

where $y$ is the true label of the training set, $f(x)$ is the predicted label, $N$ is the number of training samples, $l$ denotes the loss function, $\lambda$ is the regularisation parameter used to penalise overfitting, $P$ is the number of features and $w$ is the weight associated with each feature of the dataset.

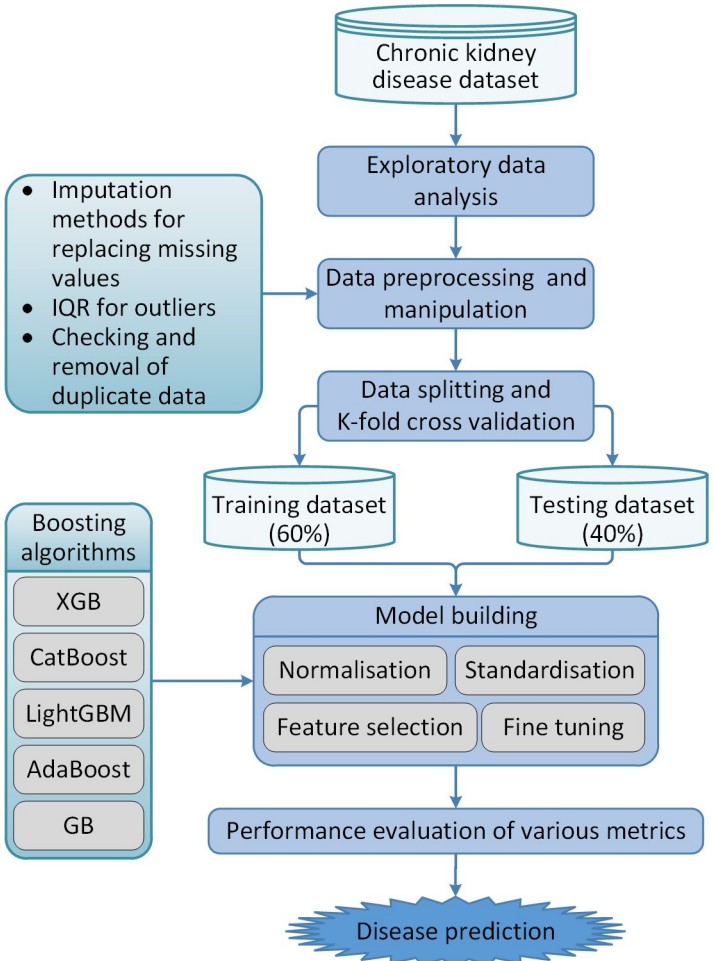

**Fig 1. The workflow of the proposed ensemble learning based CKD prediction.**

**LightGBM.** LightGBM is an extension of a gradient boosting algorithm, capable of handling large datasets with less memory utilisation during the model evaluation process [56]. Gradient-based one-side sampling method is used for splitting the data samples, reducing the number of features in sparse datasets during training. The objective function for the LightGBM algorithm is defined using Eq 3.

$$L(\theta) = \sum\nolimits_{i=1}^{N} l(y_i, \hat{y}_l) + \sum\nolimits_{i=1}^{T} \omega(f_j) \tag{3}$$

where $\theta$ is a set of model parameters, $N$ is the number of training samples, $l$ denotes the loss function, $y_i$ is the true label of $i^{th}$ sample, $\hat{y}_l$ is the predicted label for the model, $f_j$ is the $j^{th}$ decision tree, $T$ is the number of trees, and $\omega$ is the regularisation term.

**AdaBoost.** AdaBoost works by adjusting all the weights without prior knowledge of weak learners [57]. The weakness of all the base learners is measured by the estimator's error rate while training the models. Decision tree stumps are widely used with the AdaBoost algorithm to solve classification and regression problems. The objective function for the AdaBoost algorithm is defined using Eq 4.

$$L(H) = \sum\nolimits_{i=1}^{N} exp(-y_i * H(x_i)) \tag{4}$$

where $H(x_i)$ is the prediction of the classifier on the $i^{th}$ sample $x_i$ and $y_i$ is its corresponding true label in {-1, +1}and $N$ denotes the number of training samples.

**Gradient boosting.** In this method, the weak learners are trained sequentially, and all estimators are added one by one by adapting the weights [58]. The gradient boosting algorithm focuses on predicting the residual errors of previous estimators and tries to minimise the difference between the predicted and actual values. The objective function for the gradient boosting algorithm is written using Eq 5.

$$L(\theta) = \ min_F \ \sum_{i=1}^{N} l(y_i, Fmathbf(x_i)) \tag{5}$$

where $F$ is the ensemble model, $n$ is the number of training examples, $y_i$ is the true label of the $i^{th}$ sample, $l$ denotes the loss function, and $Fmathbf(x_i)$ is the output of the ensemble model on example $mathbf(x_i)$.

# 4. Dataset collection and manipulation

We used the CKD data set (https://archive.ics.uci.edu/ml/datasets/chronic_kidney_disease), publicly available at the UCI machine learning repository, for the experiment. The dataset was collected from Apollo Hospitals, Managiri, India.

## 4.1 Dataset description

The dataset contains 400 instances and 25 attributes. The first 24 attributes are predicate/independent, and the last one is a dependent/target attribute. Among the attributes, 11 are numeric, and 14 are categorical. The attributes are described in Table 1. It represents the information about considered attributes, the description of attributes, their measurements, and the range values.

Table 2 describes the attribute information with their measures like count of records, mean, standard deviation (std), minimum (min) value, and maximum (max) value. For example, the blood pressure (bp) attribute has a count value of 400, mean 76.175, std 13.769, min 50, and max 180, respectively.

## 4.2 Data preprocessing

We performed some preprocessing on the considered CKD dataset to make the dataset most usable. The purpose was to transform the available raw data into a format easily understood by the ensemble learning algorithms. We conducted the following steps as data preprocessing:

a. Identify and replace duplicate values.

b. Identify and replace missing values.

c. Detect and replace the outliers.

d. Convert categorical variables to numerical values using one-hot encoding.

e. Perform data transformation (-1 to 1) and scaling (0 to 1).

The results of the above steps are discussed below.

**Class balancing.** The training dataset should be balanced of positive and negative instances to achieve reasonable prediction. From Fig 2(A), it can be observed that the considered dataset was highly biased toward the positive class, i.e., "patients having CKD" over the negative class, "patients not having CKD." To minimise this difference, we used SMOTE to

**Table 1. Attributes information of the dataset.**

| Attribute | Description | Measurement | Value range |
|---|---|---|---|
| Age (age) | Participant's age | Years | 2–90 |
| Blood pressure (bp) | Participant's blood pressure | mm/hg | 50–180 |
| Specific gravity (sg) | Urine specific gravity of the participant | Nominal | 1.005–1.025 |
| Albumin (al) | Blood volume of the participant | Nominal | 0–5 |
| Sugar (su) | Participant's sugar level in the blood | Nominal | 0–5 |
| Red blood cells (rbc) | Normality of red blood cells of the participant | Categorical | 0 or 1 |
| Pus cell (pc) | Normality of pus cells of the participant | Categorical | 0 or 1 |
| Pus cell clumps (pcc) | Presence of pus cell clumps in the participant's urine | Categorical | 0 or 1 |
| Bacteria (ba) | Presence of bacteria in the participant's urine | Categorical | 0 or 1 |
| Blood glucose random (bgr) | Blood sugar test of the participant | mgs/dl | 22–490 |
| Blood urea (bu) | Nitrogen level in the participant's blood | mgs/dl | 1.50–391 |
| Serum creatinine (sc) | Creatinine level in the participant's blood | mgs/dl | 0.40–76 |
| Sodium (sod) | Sodium level in the participant's blood | mEq/L | 4.50–163 |
| Potassium (pot) | Potassium level in the participant's blood | mEq/L | 2.50–47 |
| Haemoglobin (hemo) | Haemoglobin measure in the participant's blood | Gms | 3.10–54 |
| Packed cell volume (pcv) | Measure and size of RBCs in the participant's blood | Numeric | 9.00–54 |
| White blood cell count (wc) | WBCs count in the participant's blood | Cells/cumm | 2200–26400 |
| Red blood cell count (rc) | RBCs count in the participant's blood | Millions/ cumm | 2.10–8 |
| Hypertension (htn) | If the participant has hypertension | Categorical | 0 or 1 |
| Diabetes mellitus (dm) | If the participant has diabetes | Categorical | 0 or 1 |
| Coronary artery disease (cad) | If the participant has coronary artery disease | Categorical | 0 or 1 |
| Appetite (appet) | Participant's desire or need for something to eat | Categorical | 0 or 1 |
| Pedal edema (pe) | If the participant has swelling in the ankles and feet | Categorical | 0 or 1 |
| Anaemia (ane) | Deficiency in RBCs of the participant | Categorical | 0 or 1 |
| Class (outcome) | If the participant has CKD | Categorical | 0 or 1 |

balance the dataset. From Fig 2(B), it can be observed that the resultant dataset is fairly balanced.

**Exploratory data analysis.** We used different data visualisation tools to visualise and analyse the distribution of the data samples. Fig 3 shows the normally distributed histograms that group all the attributes of the considered dataset within the range values. Here, the X- and Y-axes describe the input attributes, and their corresponding values, respectively. Fig 4 plots the probability density using the kernel density estimation (KDE) method. The X- and Y-axes denote each attribute's parameter value and probability density function, respectively. Fig 5 depicts the boxplot of all the considered attributes of the dataset. It provides a good indication of how the dispersion of values is spread out. To handle the outliers in the dataset, the IQR method was used.

**Correlation coefficient analysis.** To identify and plot the relationship among the dataset attributes, we used the correlation coefficient analysis (CCA) method. A strong association/relationship between the set of independent and dependent attributes indicates a good-quality dataset. Fig 6 presents the CCA of the dataset attributes used in the experiment. The relationship range lies between +1 to -1 along the X- and Y-axes.

**Data wrangling and cleaning.** To clean the dataset, we identified the missing values using the isnull() method and then calculated the percentage of null values present in the dataset. We used the data imputation methods (mean, median, fill, and original) to replace the null values. The missing values were replaced using the column's mean, median, and mode. We used

**Table 2. Attributes information of the dataset.**

| Attribute | Count | Mean | Std | Min | Max |
|---|---|---|---|---|---|
| Age (age) | 400 | 51.585 | 17.308 | 2 | 90 |
| Blood pressure (bp) | | 76.175 | 13.769 | 50 | 180 |
| Specific gravity (sg) | | 1.017 | 0.005 | 1.005 | 1.025 |
| Albumin (al) | | 1.057 | 1.343 | 0 | 5 |
| Sugar (su) | | 0.450 | 1.084 | 0 | 5 |
| Red blood cells (rbc) | | 0.727 | 0.445 | 0 | 1 |
| Pus cell (pc) | | 0.773 | 0.420 | 0 | 1 |
| Pus cell clumps (pcc) | | 0.105 | 0.307 | 0 | 1 |
| Bacteria (ba) | | 0.055 | 0.228 | 0 | 1 |
| Blood glucose random (bgr) | | 149.710 | 78.481 | 22 | 490 |
| Blood urea (bu) | | 57.426 | 49.286 | 1.500 | 391 |
| Serum creatinine (sc) | | 3.072 | 5.617 | 0.400 | 76 |
| Sodium (sod) | | 136.790 | 10.039 | 4.500 | 163 |
| Potassium (pot) | | 4.605 | 2.857 | 2.500 | 47 |
| Haemoglobin (hemo) | | 12.332 | 2.926 | 3.100 | 17.80 |
| Packed cell volume (pcv) | | 37.843 | 9.292 | 9 | 54 |
| White blood cell count (wc) | | 8448 | 2951.563 | 2200 | 26400 |
| Red blood cell count (rc) | | 4.473 | 1.009 | 2.100 | 8 |
| Hypertension (htn) | | 0.368 | 0.483 | 0 | 1 |
| Diabetes mellitus (dm) | | 0.343 | 0.475 | 0 | 1 |
| Coronary artery disease (cad) | | 0.085 | 0.279 | 0 | 1 |
| Appetite (appet) | | 0.795 | 0.404 | 0 | 1 |
| Pedal edema (pe) | | 0.190 | 0.393 | 0 | 1 |
| Anaemia (ane) | | 0.150 | 0.358 | 0 | 1 |
| Class (outcome) | | 0.625 | 0.485 | 0 | 1 |

the IRQ method to detect the outliers and replace them using the Z-score method. The Z-score method shifts the distribution of all the data samples and makes the mean 0. Using data cleaning methods, we further checked for duplicate, inconsistent, and corrupt values in the dataset and neutralised them wherever applicable.

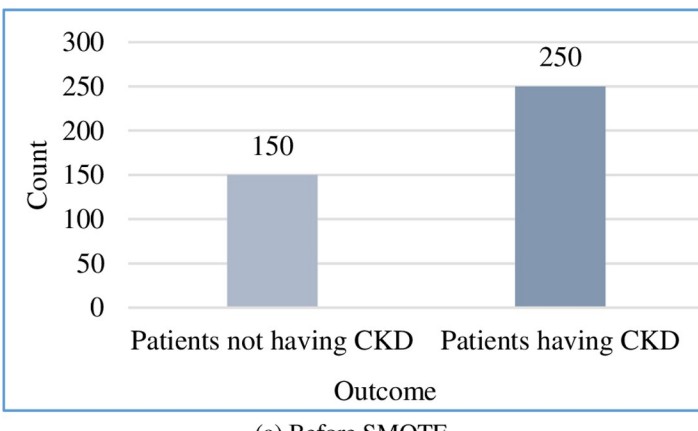

(a) Before SMOTE

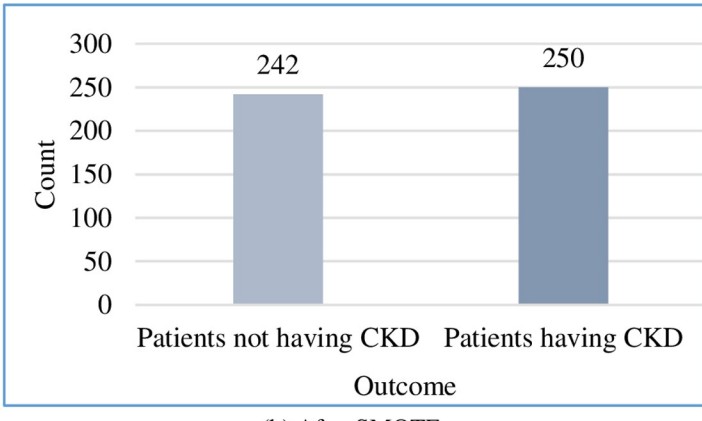

(b) After SMOTE

**Fig 2. Dataset balancing using SMOTE.**

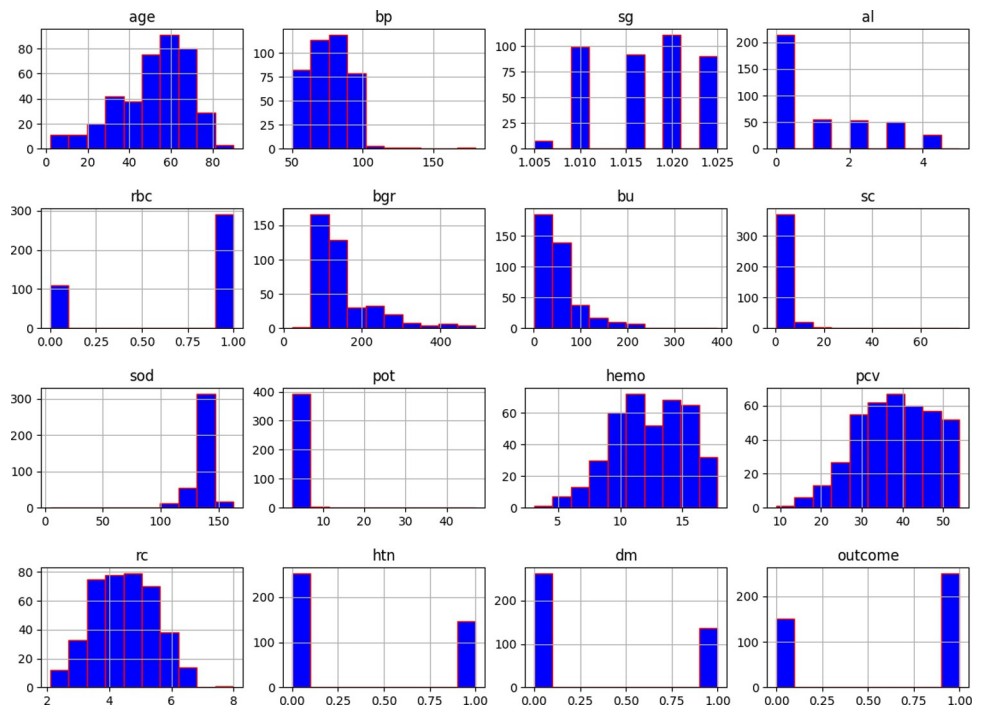

**Fig 3. Histogram of the dataset attributes.**

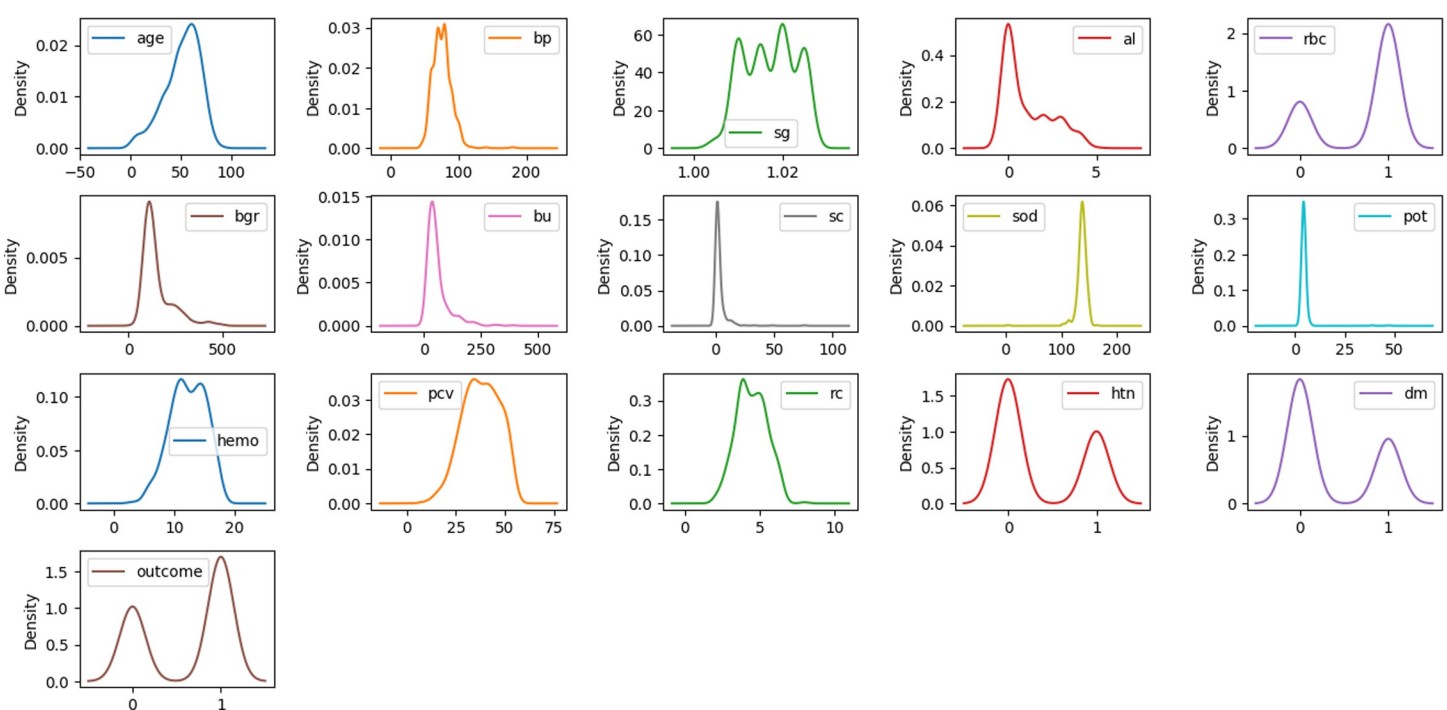

**Fig 4. Density plot of the dataset attributes.**

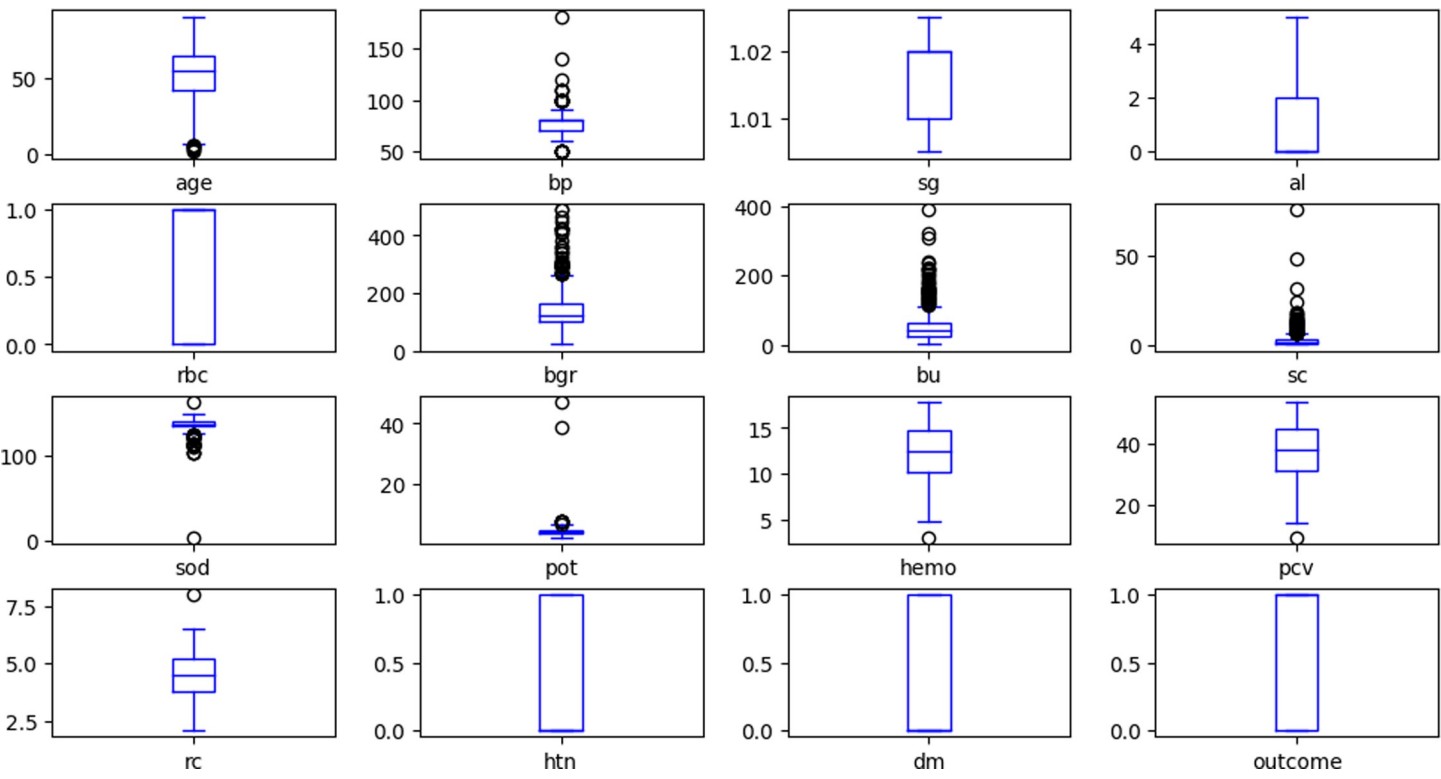

**Fig 5. Boxplot of the dataset attributes.**

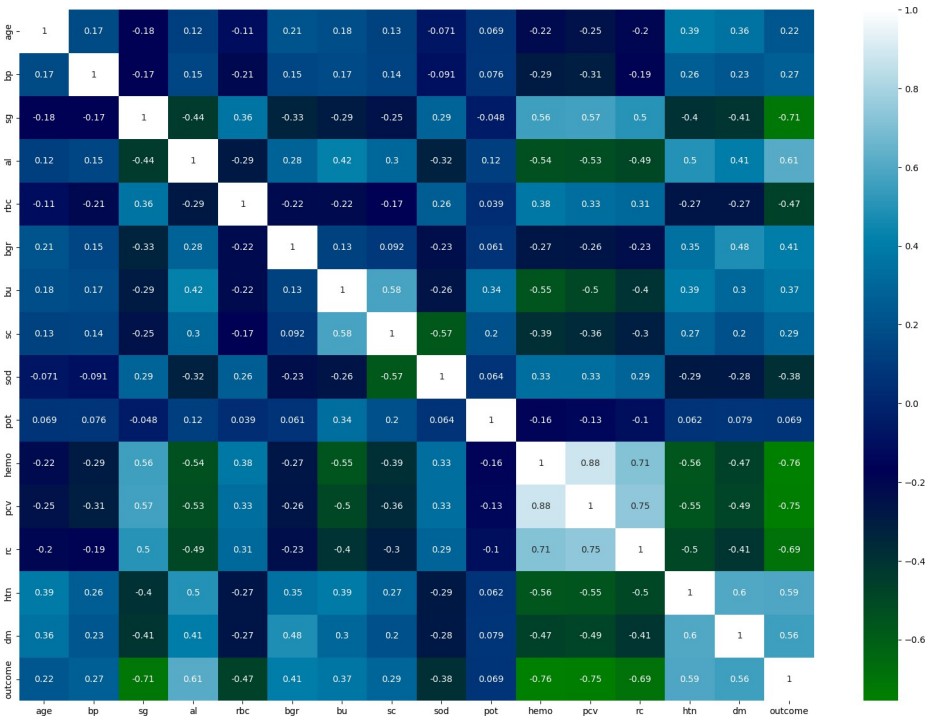

**Fig 6. Correlation coefficient analysis of the dependent and independent attributes in the dataset.**

**Data standardisation and normalisation.** We used the MinMaxScaler() for feature scaling. We scaled the data values using Eq 1 for standardisation and batch normalisation. Here, the data mean is set to 0 and the standard deviation to 6.

$$N(X) = \frac{\sum_{i=1}^{N} x_i - x_{min}}{x_{max} - x_{min}} \tag{6}$$

where $N$, $X$, $x_i$, $\sigma(x)$, $x_{min}$, and $x_{max}$ denote the total sample in the data, i[th] attribute, the mean of the attributes, the sample variance of the attributes, the minimum value of the sample, and the maximum value of the sample, respectively.

## 5. Experiment, results, and discussion

In this section, we present the experimental details of this work and the obtained results by using the five boosting algorithms to predict CKD. We used 60% of the dataset to train the boosting algorithms and the rest 40% to test and validate their efficacy. The evaluations are extensively discussed in terms of accuracy, recall, precision, F1-score, micro-weighted, average-weighted, and AUC-ROC (area under curve-receiver operating characteristic) curve for each algorithm.

### 5.1 Hardware and software specifications

An HP Z60 workstation was used to carry out this research work. The hardware specification of the system is: Intel XEON 2.4 GHz CPU (12 core), 8 GB RAM, 1 T.B. hard disk, with Windows 10 pro-64-bit O.S. environment. As software requirements, we used the GUI-based Anaconda Navigator, the web-based computing platform Jupyter notebook, and Python as the programming language.

### 5.2 Feature importance

The feature importance is used to assess the contribution of an independent/predicate attribute that leads to CKD. Generally, not all attributes contribute to disease prediction. For instance, after running all five boosting algorithms on the original dataset, we found that the attributes–'ane', 'appet', 'ba', 'cad', 'pc', 'pcc', 'pe', 'su', and 'wc' have no role in CKD prediction. Hence, we eliminated these attributes from the dataset and kept only those that contributed at least for one algorithm, as shown in Fig 7.

We used the forward selection, a wrapper method, to calculate the feature importance [59]. A higher F-score of a feature indicates more importance of an attribute. For example, in Fig 7, it can be seen that the haemoglobin (hemo) attribute has the highest contribution in the CKD prediction for all the algorithms.

### 5.3 Hyperparameter tuning

We used the grid search method for hypermeter tuning to achieve optimality in the proposed model's performance. By specifying a grid or a specified set of values for each hyperparameter, grid search enables methodically examining various combinations of hyperparameters. This ensures that all the options are tried to find the optimal values of the hyperparameters. The deterministic nature of grid search ensures consistency, i.e., it always yields the same outcomes when the same hyperparameters and data are used. This characteristic facilitates transparent testing and evaluation by making results simple to replicate and compare. One of the major advantages of grid search is that it is fairly straightforward to implement. Also, most of the machine learning frameworks and libraries provide built-in functions or modules for grid

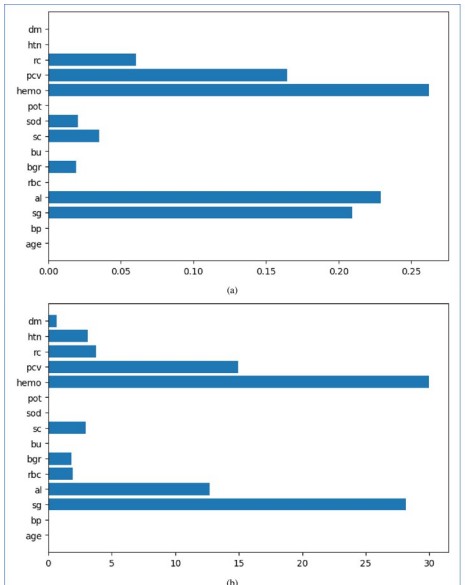
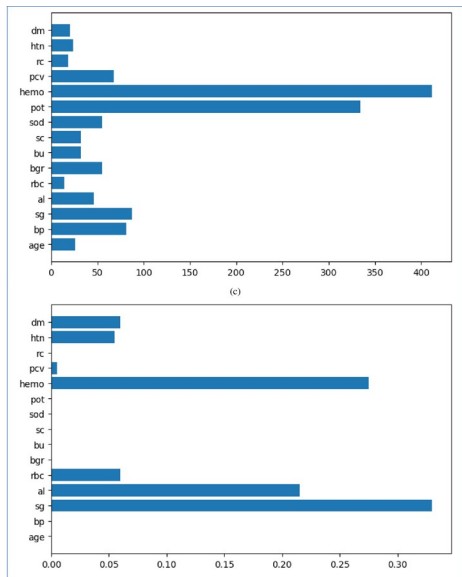
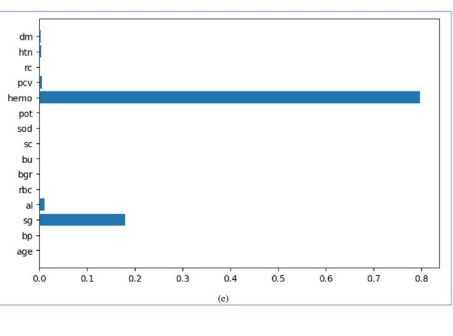

**Fig 7. Contributing features in CKD prediction for all boosting algorithms.**

search. The best values of the hyperparameters found for each algorithm are shown in Table 3. The listed values for each parameter for the respective algorithm were found to be the best performers in our experiment.

## 5.4 Cross-validation scheme

Cross-validation is conducted to provide an unbiased evaluation of the prediction model. We performed the *k*-fold cross-validation to validate the performance of the proposed model on the training dataset. Here, we kept the value of *k* as 6. Based on the validation bias, the hyperparameters used in the experiment were tuned.

## 5.5 Performance evaluation

In this section, the performance of the proposed prediction model for the considered boosting algorithms is discussed in terms of different performance metrics.

**5.5.1 Classification accuracy.** The classification performances of the algorithms are evaluated using a confusion matrix. The confusion matrices of all five boosting algorithms applied on the test dataset are shown in Fig 8. The left upper and the right lower boxes denote the

**Table 3. The optimal hyperparameters of boosting algorithms.**

| Boosting algorithm | Hyperparameters |
|---|---|
| XGBoost | XGBClassifier (learning_rate = 0.1, n_estimators = 1000, max_depth = 5, min_child_weight = 6, 'reg_alpha': 60.0, subsample = 0.6, colsample_bytree = 0.8, 'gamma': 4.20). |
| CatBoost | CatBoostClassifier (random_state = 0, learning_rate = [0.1, 0.05], n_estimators = 100, max_depth = [1,3,5], leaf_reg', 2.0, 8, 16, min_child_samples = 2, 4, 6, |
| LightGBM | LightGBM (boosting_type = 'lgbm', random_state = 45, learning_rate = 0.1, n_estimators = 1000, max_depth = 2, min_child_samples = 250, silent = True, n_jobs = 6). |
| AdaBoost | GridSearchCV (random_state = 45, learning_rate = [0.01, 0.05], n_estimators = 200, algorithm = 'SAMME.R', n_jobs = n_jobs). |
| Gradient boosting | GridSearchCV (random_state = 45, learning_rate = [0.1, 2, 5], estimators = GradientBoostingClassifier(), max_depth = 4, weight = 6, verbose = 1). |

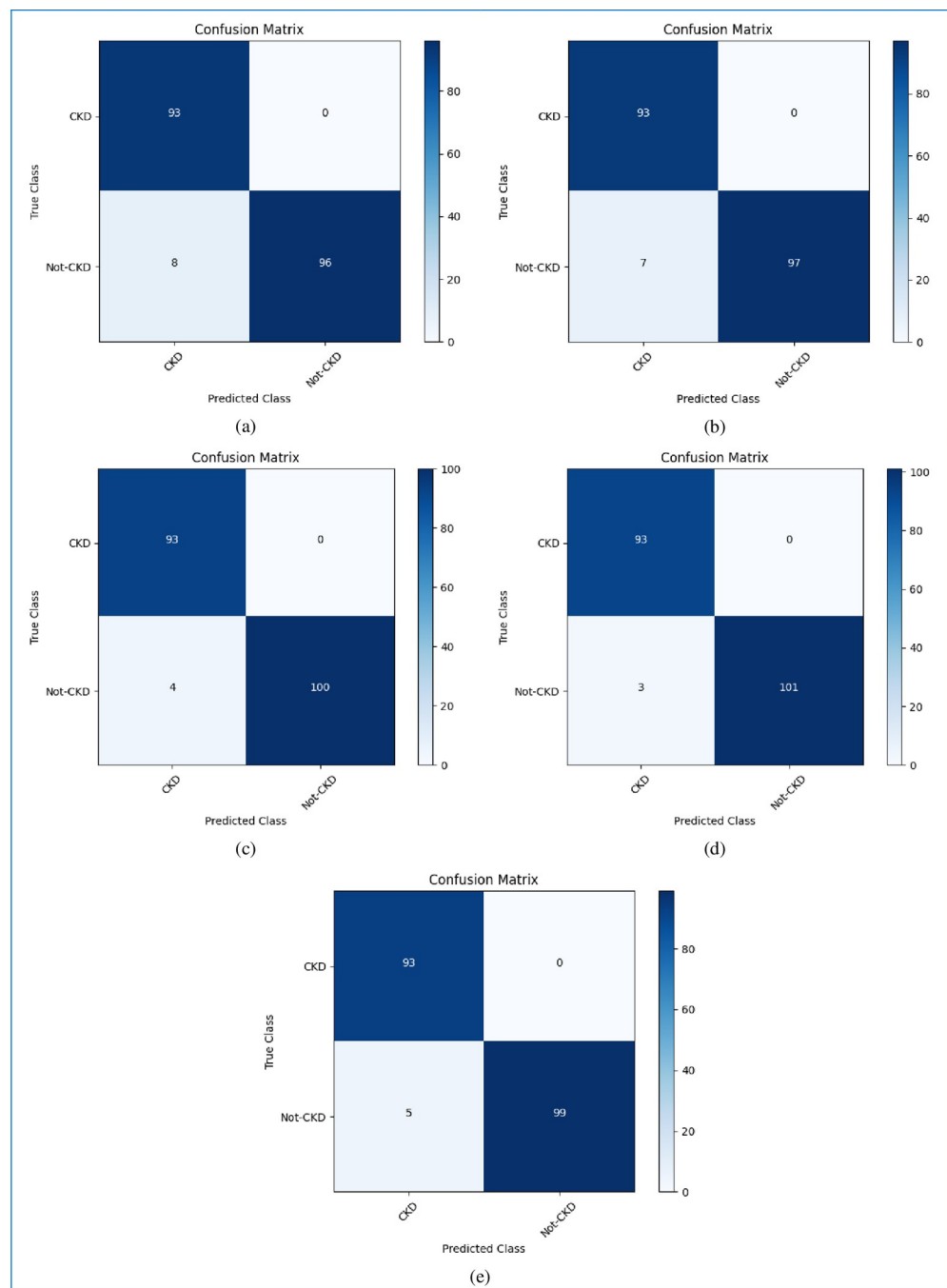

**Fig 8. Confusion matrices of the prediction performance on the test set for all the five boosting algorithms.**

correct predictions for the patients having (true positive) and not having (true negative) CKD, respectively. The right upper box and the left lower box indicate the number of incorrect predictions for patients having (false positive) and not having (false negative) CKD, respectively. The training and testing accuracies of all the boosting algorithms are shown in Fig 9. As per our experiment, on the test dataset, AdaBoost outperformed other algorithms by attaining the maximum accuracy rate for the training set of 100% and the test set of 98.47%, followed by

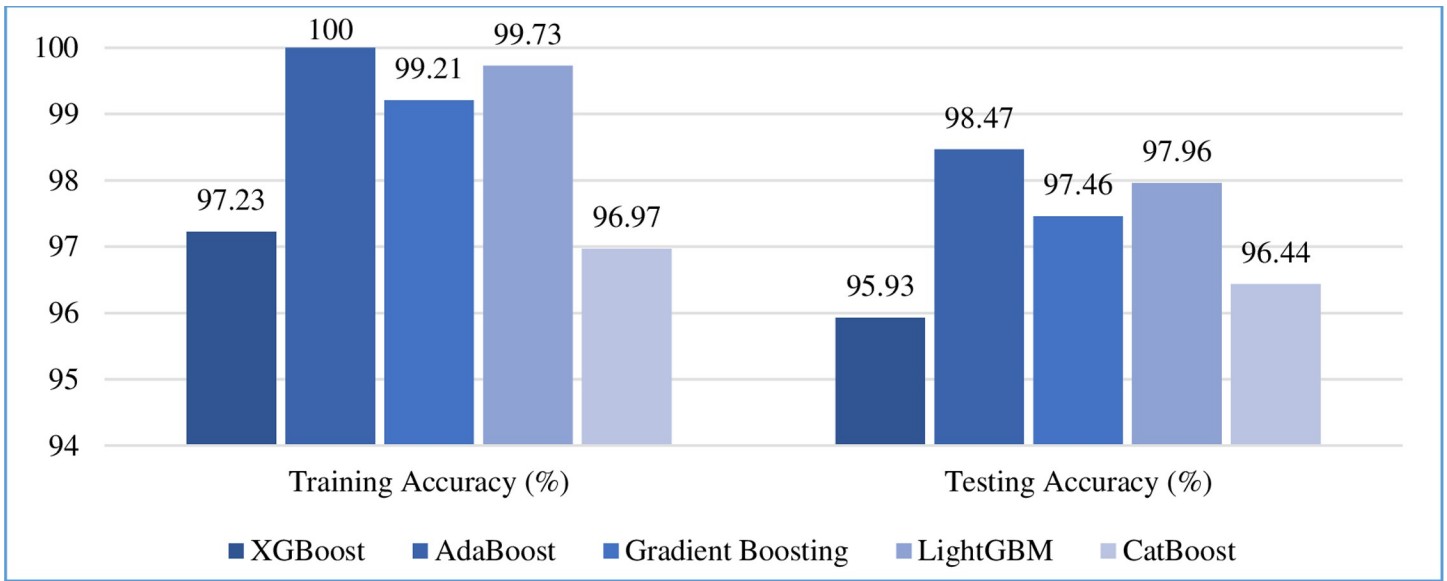

**Fig 9. Training and testing accuracy statistics of all the boosting algorithms.**

LightGBM, gradient boost, XGBoost, and CatBoost at 99.73%, 99.21%, 97.23%, and 96.97%, respectively on the training set, and 97.96%, 97.46%, 95.93%, and 96.44%, respectively on the test set.

**5.5.2 Other measurements.** In addition to accuracy, we calculated the precision, recall, F1-score, and support of the five boosting algorithms on the test set, as shown in Figs 10–13, respectively. In addition, the macro and weighted average were measured for both classes (0: no CKD, 1: CKD). As shown in those figures, AdaBoost produced the best precision in identifying the presence of CKD, while all algorithms identified the non-infection of CKD with equal precision. AdaBoost has a better recall and F1-score in confirming the absence of CKD.

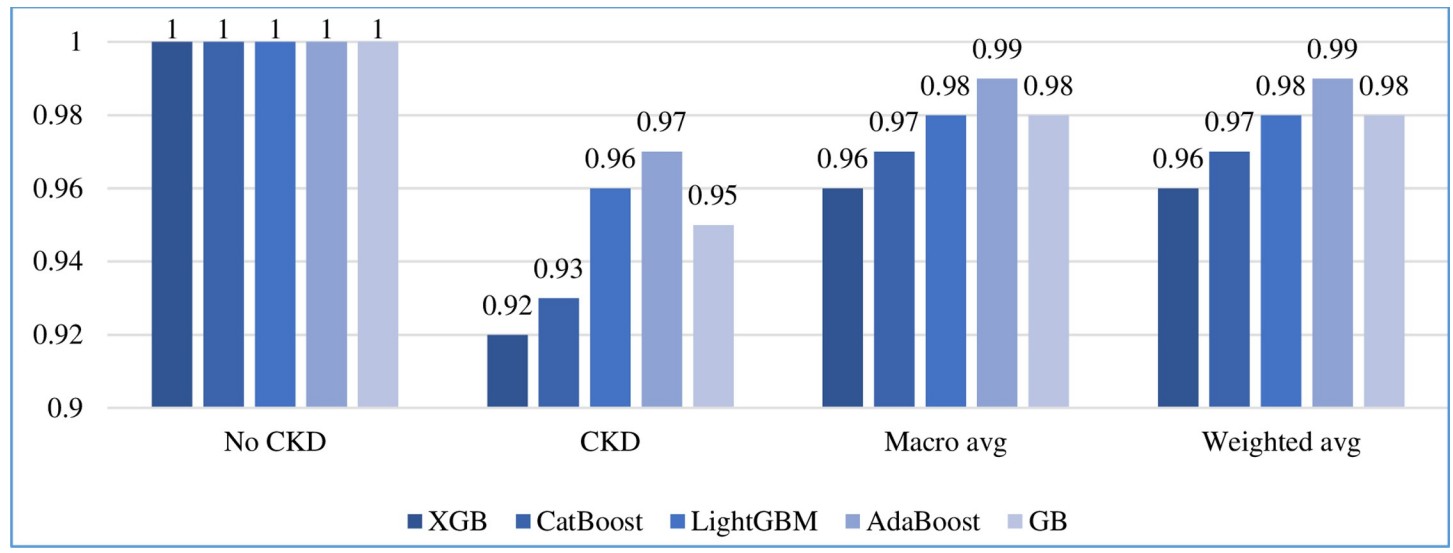

**Fig 10. Precision comparison of five boosting algorithms on test set.**

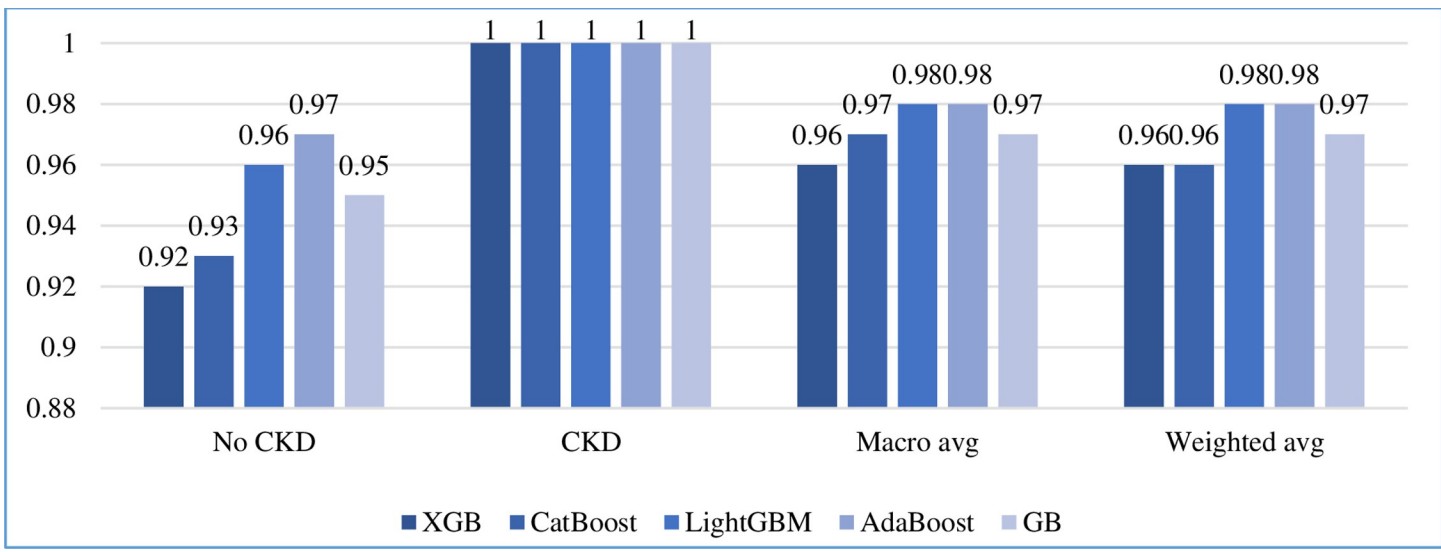

**Fig 11. Recall comparison of five boosting algorithms on test set.**

Regarding the case of support, i.e., the occurrence of class, AdaBoost performs slightly better than the other algorithms.

**5.5.3 AUC-ROC curve.** The AUC-ROC curve was used to show the prediction ability of the boosting algorithms at different thresholds. It represents a false-positive rate (FPR) vs. a true-positive rate (TPR) along the x-axis and y-axis. A larger AUC-ROC area suggests the model's ability to distinguish between 0's and 1's, leading to a better prediction. Also, an AUC value closer to 1 denotes a good separability measure, while in the case of an AUC of below 0.5, the model becomes ineffective in separating the classes, denoting the bad measure of disassociation. The AUC-ROC for the experiment is shown in Fig 14. It can be observed that AdaBoost performs best while XGBoost performs worst.

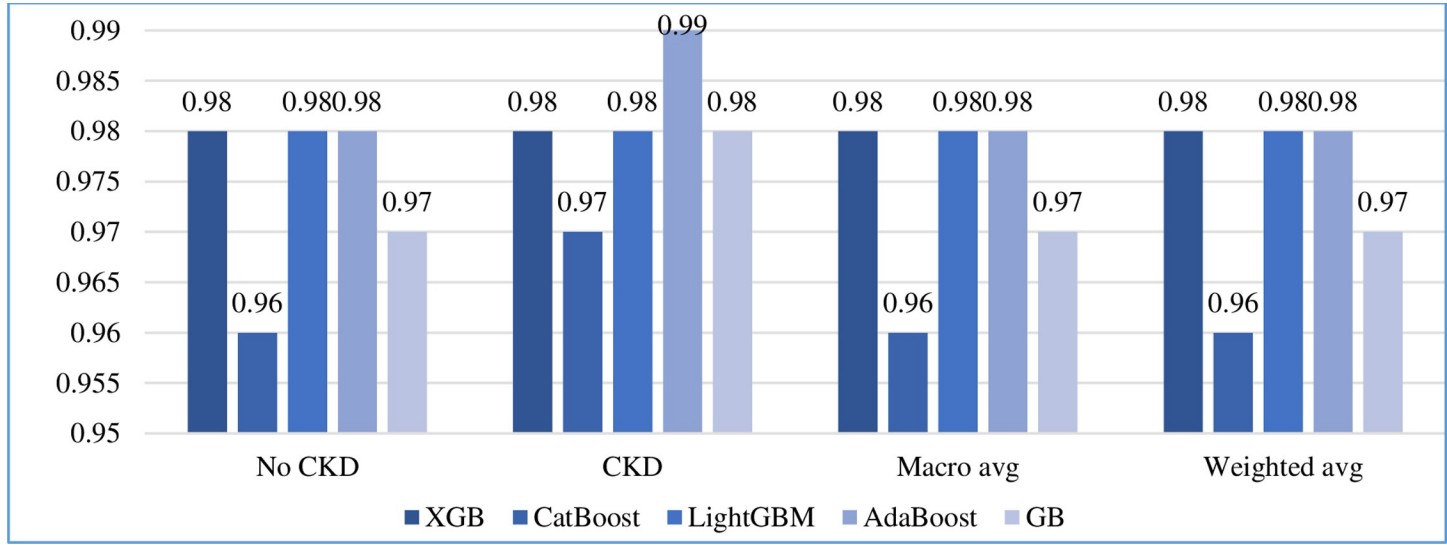

**Fig 12. F1-score comparison of five boosting algorithms on test set.**

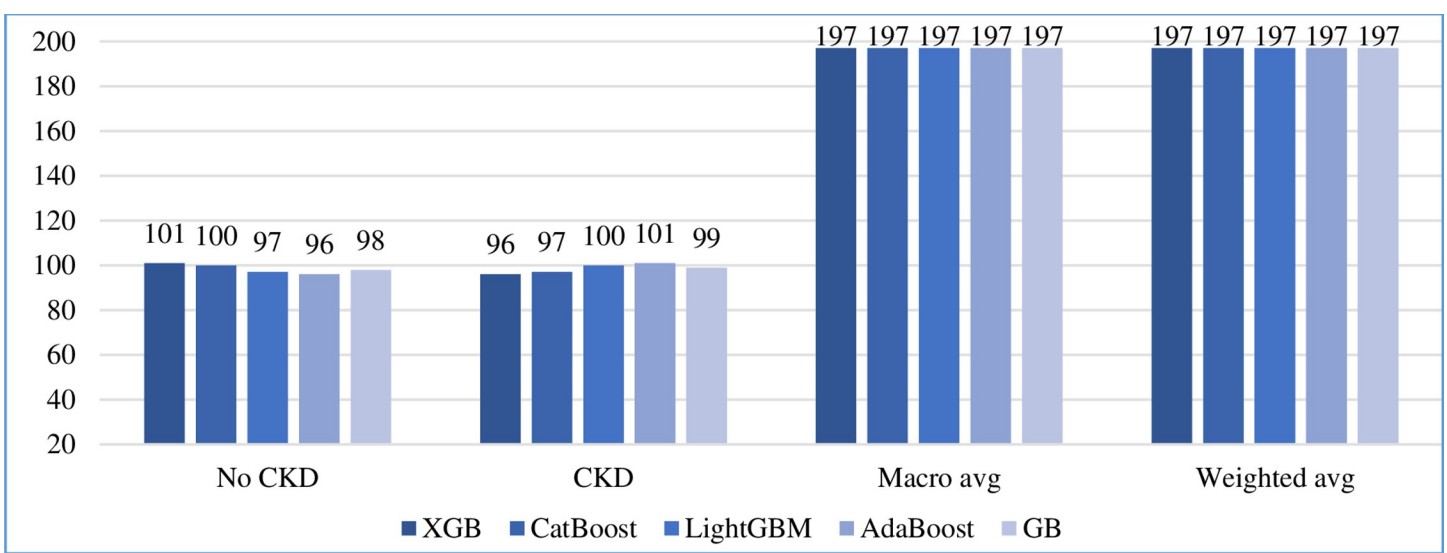

**Fig 13. Support comparison of five boosting algorithms on test set.**

## 6. Comparative analysis

Table 4 presents a comparative analysis of the five boosting algorithms applied on the test data-set in terms of accuracy, misclassification rate, and runtimes. It can be observed that AdaBoost has the highest accuracy and least misclassification rate, but it has a slightly higher runtime than LightGBM and XGBoost.

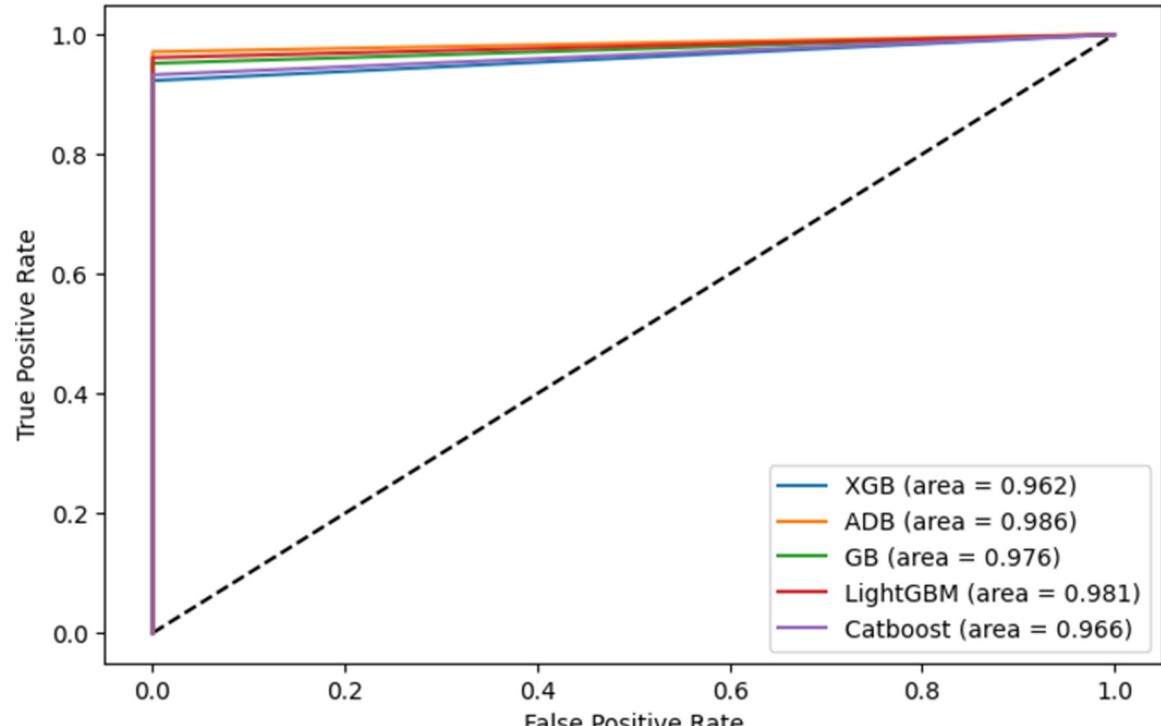

**Fig 14. AUC-ROC curves for the experimented boosting algorithms.**

**Table 4. Comparative analysis of the considered algorithms performed on the test set.**

| Algorithm | Accuracy (%) | Misclassification rate (%) | Runtime (seconds) |
|---|---|---|---|
| XGB | 95.93 | 4.07 | 1.215 |
| CatBoost | 96.44 | 3.56 | 2.009 |
| LGBM | 97.96 | 2.04 | 1.005 |
| ADB | 98.47 | 1.53 | 1.970 |
| GB | 97.46 | 2.54 | 2.752 |

Since, in our experiment, we found AdaBoost to have the best overall performance in predicting CKD, we compared it with a few related research works in terms of accuracy, as shown in Table 5. The justification for achieving higher accuracy can be credited to the adopted procedures like data imputation for handling missing values, detection and replacing outliers, and effective data standardisation and normalisation.

## 7. Conclusion, limitations, and future directions

Diagnosis and prevention of chronic kidney disease have become challenging for healthcare professionals and other concerned authorities. It can be mitigated to some extent if it can be pre-diagnosed in well advance. In this paper, we attempted to predict CKD using an ensemble learning approach. Specifically, we used five boosting algorithms: XGBoost, CatBoost, LightGBM, AdaBoost, and gradient boosting. We employed different preprocessing techniques like the imputation method for handling missing values and min-max scalar and Z-score for data standardisation and normalisation. In addition, hyperparameter techniques like grid search were used to find the optimal parameter values. Furthermore, feature selection was carried out for each algorithm. AdaBoost emerged as the overall best performer in accuracy (99.17%), precision, recall, f1-score, and support in the experiment. AdaBoost also attained better results for AUC-ROC and misclassification rate. Comparing our proposed model with similar works, we found that our method outperformed others.

**Table 5. Comparison of the proposed work with existing similar works.**

| Research work | Ensemble techniques adopted | Dataset used | Highest accuracy | Precision | Recall | AUC/ROC |
|---|---|---|---|---|---|---|
| Jongbo et al. [1] | Individual + bagging ensemble approach + random subspace ensemble (naive Bayes, kNN, and decision tree) | Chronic Kidney Dataset collected from UCI machine learning repository | 98.30% with decision tree using random subsample ensemble | - | 98.50% | 100% |
| Kumar et al. [42] | SVM, C4.5 decision tree, PSO-MLP, decision tree, and artificial bee colony C4.5 | | 92.76% with artificial bee colony 4.5 | 0.57% | 0.42% | - |
| Saurabh Pal [43] | Logistic regression, decision tree, SVM, and bagging method | | 95.92% with decision tree | 99% | 98% | - |
| Hasan and Hasan [44] | AdaBoost, bootstrap aggregating, extra trees, gradient boosting, and random forest | | 99% with AdaBoost | 98% | 100% | 99% |
| Wibawa et al. [45] | AdaBoost based on KNN | | 98.01% with AdaBoost | 97.86% | 97.83% | - |
| Ebiaredoh-Mienye et al. [46] | Logistic regression, decision tree, XGBoost, random forest, SVM, and CS AdaBoost | | 99.80% with C.S. AdaBoost | 97.50% | 100% | 98% |
| Emon et al. [47] | Logistic regression, naive Bayes, multilayer perceptron, stochastic gradient descent, adaptive boosting, bagging, decision tree, and random forest | | 99% with Random forest | 98.50% | 98.50% | 98% |
| Ramaswamyreddy et al. [48] | Tree bag, AdaBoost, gradient boosting, and random forest | | 99% with gradient boosting | - | - | - |
| Our method | XGB, CatBoost, LGBM, ADB, and gradient boosting | | 98.47% with ADB | 98.50% | 98.50% | 98.60% |

Though the proposed model performed relatively well, it has some obvious limitations. The size of the considered dataset is small, which may limit the prediction model's performance in generic situations. It is observed that most of the features are having least contribution towards CKD. A more balanced dataset would lead to a better prediction model.

As an extension of this work, other ensemble learning techniques, like bagging, stacking, etc., can be explored to improve the results. Additionally, deep learning techniques can also be experimented with the exercised dataset. To validate the effectivity of the proposed model, additional and larger datasets are needed in future. Our proposed model can be applied to other disease datasets (e.g., diabetes) with common features. We expect more powerful disease prediction models to be developed and implemented in medical diagnosis and treatment.

## Author Contributions

**Conceptualization:** Shahid Mohammad Ganie.

**Data curation:** Shahid Mohammad Ganie.

**Formal analysis:** Shahid Mohammad Ganie, Pijush Kanti Dutta Pramanik.

**Funding acquisition:** Zhongming Zhao.

**Methodology:** Shahid Mohammad Ganie.

**Software:** Shahid Mohammad Ganie.

**Supervision:** Zhongming Zhao.

**Validation:** Shahid Mohammad Ganie, Pijush Kanti Dutta Pramanik, Saurav Mallik.

**Visualization:** Shahid Mohammad Ganie, Pijush Kanti Dutta Pramanik.

**Writing – original draft:** Shahid Mohammad Ganie, Pijush Kanti Dutta Pramanik.

**Writing – review & editing:** Shahid Mohammad Ganie, Pijush Kanti Dutta Pramanik, Saurav Mallik, Zhongming Zhao.

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
