## [Decision Letter · Decision Letter 0]

24 Feb 2023

PONE-D-23-02469Chronic Kidney Disease Prediction Using Boosting Techniques based on Clinical ParametersPLOS ONE

Dear Dr. Ganie,

Thank you for submitting your manuscript to PLOS ONE. After careful consideration, we feel that it has merit but does not fully meet PLOS ONE’s publication criteria as it currently stands. Therefore, we invite you to submit a revised version of the manuscript that addresses the points raised during the review process.

We look forward to receiving your revised manuscript.

Kind regards,

Muhammad Fazal Ijaz

Academic Editor

PLOS ONE

Reviewers' comments:

Reviewer's Responses to Questions

**Comments to the Author**

1. Is the manuscript technically sound, and do the data support the conclusions?

Reviewer #1: Partly

Reviewer #2: Partly

Reviewer #3: No

2. Has the statistical analysis been performed appropriately and rigorously? 

Reviewer #1: No

Reviewer #2: I Don't Know

Reviewer #3: Yes

3. Have the authors made all data underlying the findings in their manuscript fully available?

Reviewer #1: Yes

Reviewer #2: No

Reviewer #3: No

4. Is the manuscript presented in an intelligible fashion and written in standard English?

Reviewer #1: Yes

Reviewer #2: No

Reviewer #3: No

5. Review Comments to the Author

Reviewer #1: The overall impression of the technical contribution of the current study is reasonable. However, the Authors may consider making tremendous amendments to the manuscript for better comprehensibility of the study.

1. The abstract must be re-written, focusing on the technical aspects of the proposed model, the main experimental results, and the metrics used in the evaluation. Briefly discuss how the proposed model is superior.

2. Please make sure the abbreviations are properly used. For example Area Under Curve-Receiving Operator Characteristic (AUC-ROC).

3. Additionally, method names should not be capitalized. Moreover, it is not the best practice to employ abbreviations in the abstract, they should be used when the term is introduced for the first time.

4. This paper is not a survey paper to include the references as "[30] [31] [32] [33] [34]." Please provide the significance of each of those studies and add a citation.

5. Literature must be tremendously improvised for better idea on the field of study and state of art models, authors may consider some of the relevant studies like https://doi.org/10.1371/journal.pone.0271619

6. Boosting algorithms must be adequately explain, for better idea refer and include https://doi.org/10.3390/healthcare10010085

7. How are the feature weights calculated, what was the approach or the technique that is being used.

8. Manuscript is having too many sub-sections, please minimize for better readability of the study.

9. The section Data Standardization and Normalisation must be discussed as data pre-processing phase, that has to be in background section of the manuscript.

10. what are the cases that are assumed as TP, TN, FP and FN, please explain them clearly, for better idea refer https://doi.org/10.3390/s21082852

11. Authors may present the loss functions for better comprehensibility of each of the models used in the proposed model. For better idea refer https://doi.org/10.3390/s21165386

12. Majority of the figures lack the clarity, they quality is fair but they must be explained in the text and the figures must be cited.

13. More comparative analysis with state-of-art models is desired.

14. By considering the current form of the conclusion section, it is hard to understand by PLos One Journal readers. It should be extended with new sentences about the necessity and contributions of the study by considering the authors' opinions about the experimental results derived from some other well-known objective evaluation values if it is possible.

15. English proofreading is strongly recommended for a better understanding of the study, and the quality of the figures must be tremendously improved.

Reviewer #2: This paper presents discussion on image systems for medical systems data analytics.

1. What are main aspects of novelty and advances in the described ideas?

2. What are limitations of presented approaches? How the models work in different scenarios of operation? What are weak points of presented ideas?

3. Related ideas: Deep neural network correlation learning mechanism for CT brain tumor detection, BiLSTM deep neural network model for imbalanced medical data of IoT systems, Deep learning for neurodegenerative disorder (2016 to 2022): A systematic review, .

4. Compare model to other in different operation and in different positioning of the input data.

5. What are future trends in the development of this type systems? How the development would work in different configurations? What kind of transfer and network configurations are necessary for your model?

6. How to set optimal coefficients for these models? Did you test other configurations? How were these selected?

7. Did you test the option to transform the knowledge before processing?

8. There are no comparisons to other models so we are not able to see advances of your processing.

9. Your fig 1 is not much informative since there are no details on your model thus we are not able to repeat your experiment.

10. How do you understand T in your model? Is this a time of processing or number of iterations?

Reviewer #3: In this paper, authors presented boosting techniques for chronic kidney disease prediction. However, there are some limitations that must be addressed as follows.

1. There are some typos and grammatical errors in Abstract. In addition, the abstract is not attractive. Some sentences in abstract should be modified to make it more attractive for readers.

2. In Introduction section, it is difficult to understand the novelty of the presented research work. In addition, some references are missing.

3. In related work, the existing works about patient disease prediction should be discussed: ‘A smart healthcare monitoring system for heart disease prediction based on ensemble deep learning and feature fusion’, ‘Alzheimer’s disease progression detection model based on an early fusion of cost-effective multimodal data’, Automatic detection of Alzheimer’s disease progression: An efficient information fusion approach with heterogeneous ensemble classifiers’, and ‘An intelligent healthcare monitoring framework using wearable sensors and social networking data.’

4. The authors should properly select and check the subsection title. There are so many typos and (see section 3 research methodolog.

5. The number given to each section is not correct.

6. Where are the other preprocessing steps? How is the data preprocessed?

7. What about feature selection?

8. The results are not properly discussed.

9. Captions of the Figures not self-explanatory. The caption of figures should be self-explanatory, and clearly explaining the figure. Extend the description of the mentioned figures to make them self-explanatory.

10. Equation 2 should be more clearly discussed.

11. In conclusion section, the future work should be more deeply discussed.

12. The whole manuscript should be thoroughly revised in order to improve its English.

6. PLOS authors have the option to publish the peer review history of their article (what does this mean?). If published, this will include your full peer review and any attached files.

Reviewer #1: No

Reviewer #2: No

Reviewer #3: No

---

## [Author Response · Author response to Decision Letter 0]

8 Mar 2023

Response to Reviewers’ comments

Reviewer 1 

1. The abstract must be re-written, focusing on the technical aspects of the proposed model, the main experimental results, and the metrics used in the evaluation. Briefly discuss how the proposed model is superior 

Response: We thank the reviewer for the suggestion. The abstract is written including the suggested points.

2. Please make sure the abbreviations are properly used. For example, Area Under Curve-Receiving Operator Characteristic (AUC-ROC). 

Response: We apologies for the silly mistake. It is duly corrected.

3. Additionally, method names should not be capitalized. Moreover, it is not the best practice to employ abbreviations in the abstract, they should be used when the term is introduced for the first time. 

Response: We acknowledge the reviewer’s concern regarding capitalization. However, in the literature the normal convention to write the following terms are: XGBoost, CatBoost, LightGBM, AdaBoost.

Furthermore, these terms are more popularly known by their abbreviated forms only. Writing the full forms for each of them would increase the word count of the abstract. Nevertheless, as suggested by the reviewer, the full forms of these terms are given in the text when they are used for first time.

4. This paper is not a survey paper to include the references as "[30] [31] [32] [33] [34]." Please provide the significance of each of those studies and add a citation. 

Response: We thank the reviewer for the suggestion. The goal of the sentence is to establish the importance of ensemble learning in disease diagnosis and treatment. As suggested, we re- written the sentence by including sample research works that addressed different diseases using ensemble learning.

5. Literature must be tremendously improvised for better idea on the field of study and state of art models, authors may consider some of the relevant studies like https://doi.org/10.1371/journal.pone.0271619

Response: We thank the reviewer for the suggestion. We included the suggested paper in the related work. We also improved the literature survey following the paper. A couple of recent papers are included in the related work. 

6. Boosting algorithms must be adequately explain, for better idea refer and include https://doi.org/10.3390/healthcare10010085

Response: We thank the reviewer for the suggestion. The boosting algorithms are elaborated with suitable mathematical equations.

7. How are the feature weights calculated, what was the approach or the technique that is being used. 

Response: To calculate the feature importance, we used the wrapper method. This is included in the manuscript with suitable reference.

8. Manuscript is having too many sub-sections, please minimize for better readability of the study. 

Response: We thank the reviewer for bringing our notice into this. There was a problem with the subsection 3.1 and Section 4 which should be included under Section 3. It is corrected duly. We structured the manuscript into different subsections which allowed us to express the non-overlapping contents precisely and exclusively. This would help the readers to focus and understand the fragments of the experiment in a granular way.

9. The section Data Standardization and Normalisation must be discussed as data pre-processing phase, that has to be in background section of the manuscript. 

Response: As suggested, the section Data Standardization and Normalisation is included in the Data Pre-processing section.

10. What are the cases that are assumed as TP, TN, FP and FN, please explain them clearly, for better idea refer https://doi.org/10.3390/s21082852

Response: We thank the reviewer for the suggestion. Each prediction cases are stated with respect to Figure 7.

11. Authors may present the loss functions for better comprehensibility of each of the models used in the proposed model. For better idea refer https://doi.org/10.3390/s21165386

Response: The misclassification rate of incorrect prediction is given in Table 4.

12. Majority of the figures lack the clarity, they quality is fair but they must be explained in the text and the figures must be cited. 

Response: The figures are described and cited in the text.

13. More comparative analysis with state-of-art models is desired. 

Response: As suggested by the reviewer, our work has been compared with a couple of papers in addition to the existing 6 papers, as shown in Table 5.

14. By considering the current form of the conclusion section, it is hard to understand by PLos One Journal readers. It should be extended with new sentences about the necessity and contributions of the study by considering the authors' opinions about the experimental results derived from some other well-known objective evaluation values if it is possible 

Response: The necessity and contribution of this research work is included in the Conclusion section.

15. English proofreading is strongly recommended for a better understanding of the study, and the quality of the figures must be tremendously improved. 

Response: The manuscript is proofread for possible grammatical and writing mistakes. Regarding the figures, most of them are program generated and large in size. Accommodating in a smaller scale makes a couple of figures look unclear. However, they are perfectly readable by zooming. 

Reviewer 2 

1. What are main aspects of novelty and advances in the described ideas? 

Response: In this paper, we proposed a novel CKD prediction model using ensemble learning. We used five different boosting algorithms to check the prediction performance of the model. Along with achieving better results (e.g., accuracy, precision, recall, etc.) and runtime we also assessed the contributions of all the attributes in the dataset that cause CKD.

2. What are limitations of presented approaches? How the models work in different scenarios of operation? What are weak points of presented ideas? 

Response: We thank the reviewer for the suggestion. The limitations of the work in included in the Conclusion section.

3. Related ideas: Deep neural network correlation learning mechanism for CT brain tumor detection, BiLSTM deep neural network model for imbalanced medical data of IoT systems, Deep learning for neurodegenerative disorder (2016 to 2022): A systematic review. 

Response: We thank the reviewer for the valuable suggestion. We’ve already started working on the deep learning based models for different disease predictions.

4. Compare model to other in different operation and in different positioning of the input data. 

Response: We tried with different combinations of the data preprocessing, feature selection and hyperparameter tuning and came out with the best performing model. The model was tried with five boosting algorithms. Finally, the performance of the best working combination of the proposed model with AdaBoost is compared with a number of state-of-the-art findings.

5. What are future trends in the development of this type systems? How the development would work in different configurations? What kind of transfer and network configurations are necessary for your model? 

Response: In future, more sophisticated disease prediction models will be prevalent in the medical diagnosis and treatment. The proposed model can be used for other healthcare datasets that share the commonality of features. The configuration setup will depend on the particular application requirement and the properties of the available dataset. The future works of this work is mentioned in the Conclusion section.

6. How to set optimal coefficients for these models? Did you test other configurations? How were these selected? 

Response: We tested the performance of the prediction model for combination of different coefficient values for all the tunable parameters. Among them, the optimal value sets were selected, as shown in Table 3.

7. Did you test the option to transform the knowledge before processing? 

Response: We carried out the background work to set up the base prediction model and tested it on different ensemble algorithms. Among them, in the given setup, AdaBoost performed best.

8. There are no comparisons to other models so we are not able to see advances of your processing. 

Response: The proposed model is compared with several similar published works, as shown in Table 5. It can be observed that our model outperforms the other compared works.

9. Your fig 1 is not much informative since there are no details on your model thus we are not able to repeat your experiment. 

Response: Fig. 1 shows only the overall flow of the paper. The details of each step are elaborately discussed in the manuscript.

10. How do you understand T in your model? Is this a time of processing or number of iterations? 

Response: As mentioned in Section 6, it’s the runtime of the considered algorithms on the considered dataset.

Reviewer 3 

1. There are some typos and grammatical errors in Abstract. In addition, the abstract is not attractive. Some sentences in abstract should be modified to make it more attractive for readers. 

Response: We thank the reviewer for the suggestion. The Abstract is rewritten to make it more precise and attractive.

2. In Introduction section, it is difficult to understand the novelty of the presented research work. In addition, some references are missing. 

Response: In this paper, we proposed a novel CKD prediction model using ensemble learning. We used five different boosting algorithms to check the prediction performance of the model. Along with achieving better results (e.g., accuracy, precision, recall, etc.) and runtime we also assessed the contributions of all the attributes in the dataset that cause CKD.

3. In related work, the existing works about patient disease prediction should be discussed: ‘A smart healthcare monitoring system for heart disease prediction based on ensemble deep learning and feature fusion’, ‘Alzheimer’s disease progression detection model based on an early fusion of cost-effective multimodal data’, Automatic detection of Alzheimer’s disease progression: An efficient information fusion approach with heterogeneous ensemble classifiers’, and ‘An intelligent healthcare monitoring framework using wearable sensors and social networking data.’ 

Response: We thank the reviewer for suggesting the papers. We really appreciate the works presented in the suggested papers. The papers that are most closely related to our work are cited.

4. The authors should properly select and check the subsection title. There are so many typos and (see section 3 research methodology. 

Response: We thank the reviewer for bringing our notice into this. There was a problem with the subsection 3.1 and Section 4 which should be included under Section 3. It is corrected duly. As suggested, a few sections/subsections are renamed.

5. The number given to each section is not correct. 

Response: We apologise for the unintentional mistake. The numberings are corrected.

6. Where are the other preprocessing steps? How is the data pre-processed? 

Response: The details of the data preprocessing are given in Section 4.4. The steps also pictorially shown in Fig. 1.

7. What about feature selection? 

Response: We considered all the featured in the CKD dataset. We assessed the contribution of each feature in CKD. To calculate the feature importance, we used the wrapper method. This is discussed in the manuscript in Section 5.3.3 and also shown in Fig. 13.

8. The results are not properly discussed. 

Response: The outcomes of the data preprocessing are presented in Section 4.2. The experimental results are discussed in Section 5.3. The performance of the proposed model is measured using several performance metrics such as accuracy, precision, recall, F1-score, support, AUC-ROC.

9. Captions of the Figures not self-explanatory. The caption of figures should be self-explanatory, and clearly explaining the figure. Extend the description of the mentioned figures to make them self-explanatory. 

Response: We thank the reviewer for the suggestion. 

Response: Most of the captions are rewritten for better understandability.

10. Equation 2 should be more clearly discussed. 

Response: In the original manuscript, there was no Eq. 2.

11. In conclusion section, the future work should be more deeply discussed. 

Response: As suggested by the reviewer, the Conclusion section is extended including the limitation of the work and future direction in this domain.

12. The whole manuscript should be thoroughly revised in order to improve its English. 

Response: We thank the reviewer for the suggestion. The manuscript is thoroughly checked for English and grammatical mistakes.

---

## [Decision Letter · Decision Letter 1]

3 Apr 2023

PONE-D-23-02469R1Chronic Kidney Disease Prediction Using Boosting Techniques based on Clinical ParametersPLOS ONE

Dear Dr. Ganie,

Thank you for submitting your manuscript to PLOS ONE. After careful consideration, we feel that it has merit but does not fully meet PLOS ONE’s publication criteria as it currently stands. Therefore, we invite you to submit a revised version of the manuscript that addresses the points raised during the review process.

We look forward to receiving your revised manuscript.

Kind regards,

Muhammad Fazal Ijaz

Academic Editor

PLOS ONE

Reviewers' comments:

Reviewer's Responses to Questions

**Comments to the Author**

1. If the authors have adequately addressed your comments raised in a previous round of review and you feel that this manuscript is now acceptable for publication, you may indicate that here to bypass the “Comments to the Author” section, enter your conflict of interest statement in the “Confidential to Editor” section, and submit your "Accept" recommendation.

Reviewer #1: (No Response)

Reviewer #2: (No Response)

2. Is the manuscript technically sound, and do the data support the conclusions?

Reviewer #1: Partly

Reviewer #2: No

3. Has the statistical analysis been performed appropriately and rigorously? 

Reviewer #1: Yes

Reviewer #2: I Don't Know

4. Have the authors made all data underlying the findings in their manuscript fully available?

Reviewer #1: No

Reviewer #2: No

5. Is the manuscript presented in an intelligible fashion and written in standard English?

Reviewer #1: No

Reviewer #2: No

6. Review Comments to the Author

Reviewer #1: Authors are recommended to provide more technical data as recommended to make the study evident.

1. Introduction must discuss adequately about the field of study and the limitations of the existing technologies in prediction of chronic kidney diseases.

2. Literature must be tremendously improvised my incorporating some of the relevant studies like https://doi.org/10.3390/s18072183 and https://doi.org/10.3390/diagnostics12123067

3. what addition technical contribution is made other than using the conventional classification techniques like XGBoost, CatBoost, LightGBM, AdaBoost, Gradient boosting.

4. Do the authors have the permission to re-use the data from Apollo Hospitals, Managiri, India. If yes, enclose the permission letter to use the data. (Very important as the dataset is not licensed under CC0: Public Domain.) why authors have not used PIMA dataset.

5. What Data Preprocessing techniques were performed like Normalization or scaling or anything else?

6. The architecture/block diagram of the proposed model must be presented. The notations for a few equations are not discussed in the paragraph above the equation.

7. What are the cases assumed as TP, TN, FP, FN (confusion matrix) in the current study.

8. Authors must provide the details of hyper parameters like training loss, testing loss, training accuracy, and testing accuracy.

9. What was the difference between G.B. and GB... they are interchangeable used across the results section. Check in the confusion matrix figure where G.B. is used and in Figure 8 its GB

10. Table 5 can incorporate some of the recent studies like https://doi.org/10.3390/diagnostics12112739

11. More comparative analysis was desired.

12. There are few typos in the manuscript, authors may crosscheck them.

Reviewer #2: Actually paper is the same, since Authors worked on style and text only and concerns are not solved at all thus i would suggest return to revisions.

7. PLOS authors have the option to publish the peer review history of their article (what does this mean?). If published, this will include your full peer review and any attached files.

Reviewer #1: No

Reviewer #2: No

---

## [Author Response · Author response to Decision Letter 1]

11 May 2023

Manuscript ID: PONE-D-23-02469R1

Title: Chronic Kidney Disease Prediction Using Boosting Techniques based on Clinical Parameters

We thank the two reviewers for their valuable time on evaluating our manuscript. In this second revision, we have tried our best to address and incorporate these valuable comments. The changes were highlighted by yellow in the revised manuscript.

Response to Reviewer #1

Reviewer #1: Authors are recommended to provide more technical data as recommended to make the study evident.

1. Introduction must discuss adequately about the field of study and the limitations of the existing technologies in prediction of chronic kidney diseases.

Response: We thank the reviewer for this valuable suggestion. In this second revision, the Introduction part is considerably updated as per suggestion. The limitations of existing machine learning algorithms are listed (fourth paragraph). Also, the purpose of our paper is specifically mentioned (last paragraph of Introduction section).

2. Literature must be tremendously improvised my incorporating some of the relevant studies like https://doi.org/10.3390/s18072183 and https://doi.org/10.3390/diagnostics12123067

Response: We appreciate the suggestion by the reviewer. While we felt these two papers either address different problems or their underlying methodology or approaches are completely unrelated. We’ve already incorporated most of the credible papers related to CKD and ensemble learning.

3. what addition technical contribution is made other than using the conventional classification techniques like XGBoost, CatBoost, LightGBM, AdaBoost, Gradient boosting.

Response: We thanks the reviewer for this critical point. In our work, we designed a novel CKD prediction model that includes comprehensive data preprocessing, hyper parameter selection and tuning, feature selection and estimation of feature importance. These cumulatively allowed us to achieve better performance than other prediction models (such as using AdaBoost).

4. Do the authors have the permission to re-use the data from Apollo Hospitals, Managiri, India. If yes, enclose the permission letter to use the data. (Very important as the dataset is not licensed under CC0: Public Domain.) why authors have not used PIMA dataset.

Response: We appreciate the reviewer’s concern on the permission to re-use the considered dataset. We would like to reassure that the dataset is not copyrighted and it is openly available on Kaggle and UCI machine learning repository.

The PIMA dataset comprises the records of diabetes patients. It is not for CKD and hence we did not use this dataset. We respectfully request the reviewer not to ask us to change our research topic in this manuscript, but we are surely happy to extend our work to other disease like diabetes.

5. What Data Preprocessing techniques were performed like Normalization or scaling or anything else?

Response: We thank the reviewer for raising this point. The steps for data preprocessing are included in Section 4.2. In brief, we conducted the following steps as data preprocessing: 

a) Identify and replace duplicate values. 

b) Identify and replace missing values. 

c) Detect and replace the outliers. 

d) Convert categorical variables to numerical values using one-hot encoding. 

e) Perform data transformation (-1 to 1) and scaling (0 to 1).

6. The architecture/block diagram of the proposed model must be presented. The notations for a few equations are not discussed in the paragraph above the equation.

Response: We thank the reviewer’s point on diagram presentation of our model. The block diagram of the proposed work is given in Fig. 1. We thank the reviewer for pointing the issue of the equation notations. They are duly corrected. We hope the newly revised manuscript satisfies your evaluation. 

7. What are the cases assumed as TP, TN, FP, FN (confusion matrix) in the current study.

Response: Thanks for your comment on clarification of these terms. The cases for TP, TN, FP, FN are described in Section 5.3.1. Specifically, as per Fig. 7. The left upper and the right lower boxes denote the number of correct predictions for the patients having (true positive) and not having (true negative) CKD, respectively. The right upper box and the left lower box indicates the number of incorrect predictions for patients having (false positive) and not having (false negative) CKD, respectively.

8. Authors must provide the details of hyper parameters like training loss, testing loss, training accuracy, and testing accuracy.

Response: The details of the hyperparameters are now given in Table 3. Instead of training and testing loss we presented the misclassification rate in Table 4 along with the model accuracy.

9. What was the difference between G.B. and GB… they are interchangeable used across the results section. Check in the confusion matrix figure where G.B. is used and in Figure 8 its GB

Response: We thank the reviewer for pointing out this inconsistency use of the abbreviation. In the second revised manuscript, the GB and other abbreviations are written uniformly throughout the manuscript. 

10. Table 5 can incorporate some of the recent studies like https://doi.org/10.3390/diagnostics12112739

Response: Table 5 already includes the available related studies. While we much appreciate this reviewer for valuable suggestion, unfortunately, the suggested paper (AAL and Internet of Medical Things for Monitoring Type-2 Diabetic Patients) is not related to CKD prediction in our work; hence we could not incorporate such studies in Table 5. Please note that we have already cited some other papers suggested by you/or other reviewers in the previous revision, and our manuscript is already long. Thank you again for your kind understanding of scientific merits regarding the scope and integrity. 

11. More comparative analysis was desired.

Response: In the previous revision, in response to your comment, we have performed a couple of studies as comparative analysis. While there are many computational works in the biomedical research, it is better for us to focus on CKD prediction. That says, we could not find any more good papers that used Boosting algorithm for CKD prediction. In addition, for some papers we identified in this topic, they were published in the predatory journals or low-level conferences; hence, we did not include them for comparison. We sincerely hope that the reviewer is satisfied with our overall work, and happy to do more if indeed this is required after this petition.

12. There are few typos in the manuscript, authors may crosscheck them.

Response: We thank the reviewer for carefully reading our manuscript. The manuscript is thoroughly rechecked for the typos and grammatical errors. We will also try to work with the typesetters in the proofs stage.

Reviewer #2: Actually paper is the same, since Authors worked on style and text only and concerns are not solved at all thus i would suggest return to revisions.

Response: We thank the reviewer 2 for evaluating our work as a peer reviewer. While we should have done better work in explaining our revision in the previous round, we respectfully disagree with this reviewer’s judgment on our previous revision. We hope there wasn’t any error (e.g., the response letter was not shown) in the manuscript system to this reviewer. In the previous revision, we tried to address every concern of all the three reviewers as practically as possible. The previous responses to reviewers’ comments are given below for your reference. If this reviewer still has the same concern or did not see our response letter, please let us know through editor/editorial office, so that such a misunderstanding would be avoided. Thank you again for your kind service in peer-review system. 

Reviewer Reviewer’s concerns (first review) Response/action taken

Reviewer 1 1. The abstract must be re-written, focusing on the technical aspects of the proposed model, the main experimental results, and the metrics used in the evaluation. Briefly discuss how the proposed model is superior We thank the reviewer for the suggestion. The abstract is written including the suggested points.

 2. Please make sure the abbreviations are properly used. For example, Area Under Curve-Receiving Operator Characteristic (AUC-ROC). We apologise for the silly mistake. It is duly corrected.

 3. Additionally, method names should not be capitalized. Moreover, it is not the best practice to employ abbreviations in the abstract, they should be used when the term is introduced for the first time. We acknowledge the reviewer’s concern regarding capitalisation. However, in the literature the normal convention to write the following terms are:

XGBoost, CatBoost, LightGBM, AdaBoost.

Furthermore, these terms are more popularly known by their abbreviated forms only. Writing the full forms for each of them would increase the word count of the abstract. Nevertheless, as suggested by the reviewer, the full forms of these terms are given in the text when they are used for first time.

 4. This paper is not a survey paper to include the references as "[30] [31] [32] [33] [34]." Please provide the significance of each of those studies and add a citation. We thank the reviewer for the suggestion. The goal of the sentence is to establish the importance of ensemble learning in disease diagnosis and treatment. As suggested, we re- written the sentence by including sample research works that addressed different diseases using ensemble learning.

 5. Literature must be tremendously improvised for better idea on the field of study and state of art models, authors may consider some of the relevant studies like https://doi.org/10.1371/journal.pone.0271619 We thank the reviewer for the suggestion. We included the suggested paper in the related work. We also improved the literature survey following the paper. A couple of recent papers are included in the related work. 

 6. Boosting algorithms must be adequately explain, for better idea refer and include https://doi.org/10.3390/healthcare10010085 We thank the reviewer for the suggestion. The boosting algorithms are elaborated with suitable mathematical equations.

 7. How are the feature weights calculated, what was the approach or the technique that is being used. To calculate the feature importance, we used the wrapper method. This is included in the manuscript with suitable reference.

 8. Manuscript is having too many sub-sections, please minimize for better readability of the study. We thank the reviewer for bringing our notice into this. There was a problem with the subsection 3.1 and Section 4 which should be included under Section 3. It is corrected duly. 

We structured the manuscript into different subsections which allowed us to express the non-overlapping contents precisely and exclusively. This would help the readers to focus and understand the fragments of the experiment in a granular way.

 9. The section Data Standardization and Normalisation must be discussed as data pre-processing phase, that has to be in background section of the manuscript. As suggested, the section Data Standardization and Normalisation is included in the Data Pre-processing section.

 10. What are the cases that are assumed as TP, TN, FP and FN, please explain them clearly, for better idea refer https://doi.org/10.3390/s21082852 We thank the reviewer for the suggestion. Each prediction cases are stated with respect to Figure 7.

 11. Authors may present the loss functions for better comprehensibility of each of the models used in the proposed model. For better idea refer https://doi.org/10.3390/s21165386 The misclassification rate of incorrect prediction is given in Table 4.

 12. Majority of the figures lack the clarity, they quality is fair but they must be explained in the text and the figures must be cited. The figures are described and cited in the text.

 13. More comparative analysis with state-of-art models is desired. As suggested by the reviewer, our work has been compared with a couple of papers in addition to the existing 6 papers, as shown in Table 5.

 14. By considering the current form of the conclusion section, it is hard to understand by PLos One Journal readers. It should be extended with new sentences about the necessity and contributions of the study by considering the authors' opinions about the experimental results derived from some other well-known objective evaluation values if it is possible The necessity and contribution of this research work is included in the Conclusion section.

 15. English proofreading is strongly recommended for a better understanding of the study, and the quality of the figures must be tremendously improved. The manuscript is proofread for possible grammatical and writing mistakes. 

Regarding the figures, most of them are program generated and large in size. Accommodating in a smaller scale makes a couple of figures look unclear. However, they are perfectly readable by zooming. 

Reviewer 2 1. What are main aspects of novelty and advances in the described ideas? In this paper, we proposed a novel CKD prediction model using ensemble learning. We used five different boosting algorithms to check the prediction performance of the model. Along with achieving better results (e.g., accuracy, precision, recall, etc.) and runtime we also assessed the contributions of all the attributes in the dataset that cause CKD.

 2. What are limitations of presented approaches? How the models work in different scenarios of operation? What are weak points of presented ideas? We thank the reviewer for the suggestion. The limitations of the work in included in the Conclusion section.

 3. Related ideas: Deep neural network correlation learning mechanism for CT brain tumor detection, BiLSTM deep neural network model for imbalanced medical data of IoT systems, Deep learning for neurodegenerative disorder (2016 to 2022): A systematic review. We thank the reviewer for the valuable suggestion. We’ve already started working on the deep learning based models for different disease predictions.

 4. Compare model to other in different operation and in different positioning of the input data. We tried with different combinations of the data preprocessing, feature selection and hyperparameter tuning and came out with the best performing model. The model was tried with five boosting algorithms. Finally, the performance of the best working combination of the proposed model with AdaBoost is compared with a number of state-of-the-art findings.

 5. What are future trends in the development of this type systems? How the development would work in different configurations? What kind of transfer and network configurations are necessary for your model? In future, more sophisticated disease prediction models will be prevalent in the medical diagnosis and treatment.

The proposed model can be used for other healthcare datasets that share the commonality of features. 

The configuration setup will depend on the particular application requirement and the properties of the available dataset.

The future works of this work is mentioned in the Conclusion section.

 6. How to set optimal coefficients for these models? Did you test other configurations? How were these selected? We tested the performance of the prediction model for combination of different coefficient values for all the tuneable parameters. Among them, the optimal value sets were selected, as shown in Table 3.

 7. Did you test the option to transform the knowledge before processing? We carried out the background work to set up the base prediction model and tested it on different ensemble algorithms. Among them, in the given setup, AdaBoost performed best.

 8. There are no comparisons to other models so we are not able to see advances of your processing. The proposed model is compared with several similar published works, as shown in Table 5. It can be observed that our model outperforms the other compared works.

 9. Your fig 1 is not much informative since there are no details on your model thus we are not able to repeat your experiment. Fig. 1 shows only the overall flow of the paper. The details of each step are elaborately discussed in the manuscript.

 10. How do you understand T in your model? Is this a time of processing or number of iterations? As mentioned in Section 6, it’s the runtime of the considered algorithms on the considered dataset.

Reviewer 3 1. There are some typos and grammatical errors in Abstract. In addition, the abstract is not attractive. Some sentences in abstract should be modified to make it more attractive for readers. We thank the reviewer for the suggestion. The Abstract is rewritten to make it more precise and attractive.

 2. In Introduction section, it is difficult to understand the novelty of the presented research work. In addition, some references are missing. In this paper, we proposed a novel CKD prediction model using ensemble learning. We used five different boosting algorithms to check the prediction performance of the model. Along with achieving better results (e.g., accuracy, precision, recall, etc.) and runtime we also assessed the contributions of all the attributes in the dataset that cause CKD.

 3. In related work, the existing works about patient disease prediction should be discussed: ‘A smart healthcare monitoring system for heart disease prediction based on ensemble deep learning and feature fusion’, ‘Alzheimer’s disease progression detection model based on an early fusion of cost-effective multimodal data’, Automatic detection of Alzheimer’s disease progression: An efficient information fusion approach with heterogeneous ensemble classifiers’, and ‘An intelligent healthcare monitoring framework using wearable sensors and social networking data.’ We thank the reviewer for suggesting the papers. We really appreciate the works presented in the suggested papers. The papers that are most closely related to our work are cited.

 4. The authors should properly select and check the subsection title. There are so many typos and (see section 3 research methodology. We thank the reviewer for bringing our notice into this. There was a problem with the subsection 3.1 and Section 4 which should be included under Section 3. It is corrected duly. As suggested, a few sections/subsections are renamed.

 5. The number given to each section is not correct. We apologise for the unintentional mistake. The numberings are corrected.

 6. Where are the other preprocessing steps? How is the data pre-processed? The details of the data preprocessing are given in Section 4.4. The steps also pictorially shown in Fig. 1.

 7. What about feature selection? We considered all the featured in the CKD dataset. We assessed the contribution of each feature in CKD. To calculate the feature importance, we used the wrapper method. This is discussed in the manuscript in Section 5.3.3 and also shown in Fig. 13.

 8. The results are not properly discussed. The outcomes of the data preprocessing are presented in Section 4.2. The experimental results are discussed in Section 5.3. The performance of the proposed model is measured using several performance metrics such as accuracy, precision, recall, F1-score, support, AUC-ROC.

 9. Captions of the Figures not self-explanatory. The caption of figures should be self-explanatory, and clearly explaining the figure. Extend the description of the mentioned figures to make them self-explanatory. We thank the reviewer for the suggestion. Most of the captions are rewritten for better understandability.

 10. Equation 2 should be more clearly discussed. There was no Eq. 2 in our original manuscript (first submission). If the reviewer refers to other equation, please specify and we will add some more detail, or check for this issue. 

 11. In conclusion section, the future work should be more deeply discussed. As suggested by the reviewer, the Conclusion section is extended including the limitation of the work and future direction in this domain.

 12. The whole manuscript should be thoroughly revised in order to improve its English. We thank the reviewer for the suggestion. The manuscript is thoroughly checked for English and grammatical mistakes.

---

## [Decision Letter · Decision Letter 2]

7 Jul 2023

PONE-D-23-02469R2Chronic Kidney Disease Prediction Using Boosting Techniques based on Clinical ParametersPLOS ONE

Dear Dr. Ganie,

Thank you for submitting your manuscript to PLOS ONE. After careful consideration, we feel that it has merit but does not fully meet PLOS ONE’s publication criteria as it currently stands. Therefore, we invite you to submit a revised version of the manuscript that addresses the points raised during the review process.

We look forward to receiving your revised manuscript.

Kind regards,

Anwar P.P. Abdul Majeed

Academic Editor

PLOS ONE

Reviewers' comments:

Reviewer's Responses to Questions

**Comments to the Author**

1. If the authors have adequately addressed your comments raised in a previous round of review and you feel that this manuscript is now acceptable for publication, you may indicate that here to bypass the “Comments to the Author” section, enter your conflict of interest statement in the “Confidential to Editor” section, and submit your "Accept" recommendation.

Reviewer #1: All comments have been addressed

Reviewer #2: (No Response)

Reviewer #4: (No Response)

2. Is the manuscript technically sound, and do the data support the conclusions?

Reviewer #1: Partly

Reviewer #2: No

Reviewer #4: Partly

3. Has the statistical analysis been performed appropriately and rigorously? 

Reviewer #1: Yes

Reviewer #2: No

Reviewer #4: Yes

4. Have the authors made all data underlying the findings in their manuscript fully available?

Reviewer #1: Yes

Reviewer #2: No

Reviewer #4: Yes

5. Is the manuscript presented in an intelligible fashion and written in standard English?

Reviewer #1: Yes

Reviewer #2: No

Reviewer #4: Yes

6. Review Comments to the Author

Reviewer #1: The authors have addressed all the recommendations of the reviewers in a reasonable manner, manuscript in the current from may be considered for the further phase of the editorial process.

-Captions of the Figures not self-explanatory. The caption of figures should be self-explanatory, and clearly explaining the figure. Extend the description of the mentioned figures to make them self-explanatory.

- The corresponding code may be enclosed to make the study evident.

Reviewer #2: Papers is presented in the same way as visible even after presented changes, thus i confirm my opinion from previous round

Reviewer #4: Dear Authors,

Hereby is my initial comments:

1. The dataset used is a public dataset. You need to compare related works within the similar dataset instead of comparing with other datasets. To enhance your novelty.

2. The dataset is an imbalance dataset with ratio 150:250 for with/without CKD. In my opinion, you should make sure the dataset is imbalanced or else you need to use a proper performance measure for the imbalanced dataset. By having an imbalanced dataset, your results are biased results! See your results, CKD and non CKD have big gaps.

3. Did you select important features before doing hyperparameters tuning? How many features are discarded and how many features are retained? If so, the explanation of feature importance should come before hyperparameters tuning.

4. In standard ML procedure, important features are selected to be fed into the classifiers to improve the accuracy of the model.

5. The wrapper method is used for feature importance. What type of wrapper method is used, Forward selection, Backward elimination, or Bi-directional elimination (Stepwise Selection)? Kindly explain.

6. What are the initial parameters for each boosting algorithm before doing hyperparameters tuning? Table 3 is a found best parameters not hyperparameter tuning. Revise the table caption.

7. Which dataset is used for hyperparameters tuning? 60% of the train set? What cross validation is used for this tuning?

8. What method is used for hyperparameters tuning? Grid search method for all algorithms? Kindly include the explanation of the grid search method.

9. The accuracy results should have both for train and test set. Figure 8 is a result for which set? The obtained confusion matrix in Figure 7 is for the test set?

10. Kindly explain what happened in this stage c) Detect and replace the outliers. Why did you replace the outlier? The outlier is your real data.

7. PLOS authors have the option to publish the peer review history of their article (what does this mean?). If published, this will include your full peer review and any attached files.

Reviewer #1: No

Reviewer #2: No

Reviewer #4: No

---

## [Author Response · Author response to Decision Letter 2]

24 Jul 2023

Response to Reviewers’ comments

Reviewer #1: The authors have addressed all the recommendations of the reviewers in a reasonable manner, manuscript in the current from may be considered for the further phase of the editorial process.

-Captions of the Figures not self-explanatory. The caption of figures should be self-explanatory, and clearly explaining the figure. Extend the description of the mentioned figures to make them self-explanatory.

Response: We thank the reviewer for the suggestion. We renamed the figures and tables where required.

- The corresponding code may be enclosed to make the study evident.

Response: We agree with the reviewer’s suggestion. We plan to compile the code onto an open-access repository such as GitHub or our website soon.

Reviewer #2: Papers is presented in the same way as visible even after presented changes, thus i confirm my opinion from previous round

Response: In our communication with the editor, we will not need to respond this reviewer’s criticism because he/she continuously ignored what we revised our previous versions of manuscript in response to his/her comments.

Reviewer #4: Dear Authors, hereby is my initial comments:

1. The dataset used is a public dataset. You need to compare related works within the similar dataset instead of comparing with other datasets. To enhance your novelty.

Response: We appreciate the suggestion made by the reviewer. We compared our method with similar work that used the same dataset, i.e., the Chronic Kidney Dataset collected from the UCI machine learning repository. The comparison summary is given in Table 5. In the related work section, to discuss the state-of-the-art on chronic kidney disease prediction, we needed to explore all the recent work on the same topic. 

2. The dataset is an imbalance dataset with ratio 150:250 for with/without CKD. In my opinion, you should make sure the dataset is imbalanced or else you need to use a proper performance measure for the imbalanced dataset. By having an imbalanced dataset, your results are biased results! See your results, CKD and non CKD have big gaps.

Response: We sincerely thank the reviewer for highlighting this omission. We’ve reworked with the dataset to make it balanced, as shown in Fig. 2. The whole experiment is conducted on this updated balanced dataset, ensuring that there are no biases in the outcome.

3. Did you select important features before doing hyperparameters tuning? How many features are discarded and how many features are retained? If so, the explanation of feature importance should come before hyperparameters tuning.

Response: We thank the reviewer for raising this query. In the revised manuscript, we eliminated the insignificant/non-contributing features from the dataset after finding the important features. Altogether nine attributes (‘ane’, ‘appet’, ‘ba’, ‘cad’, ‘pc’, ‘pcc’, ‘pe’, ‘su’, and ‘wc’) were discarded. We repeated the experiment with the modified dataset, the details of which are discussed in sections 5.2 to 5.5.

4. In standard ML procedure, important features are selected to be fed into the classifiers to improve the accuracy of the model.

Response: The reviewer has rightly stated that feeding the important features into the prediction model improves its performance. Complying with this suggestion, we repeated the experiment with the modified dataset (after eliminating the non-contributing features), the details of which is discussed in sections 5.2 to 5.5.

5. The wrapper method is used for feature importance. What type of wrapper method is used, Forward selection, Backward elimination, or Bi-directional elimination (Stepwise Selection)? Kindly explain.

Response: We thank the reviewer for mentioning this missed out point. We used forward selection as the wrapper method. It is now mentioned in the manuscript.

6. What are the initial parameters for each boosting algorithm before doing hyperparameters tuning? Table 3 is a found best parameters not hyperparameter tuning. Revise the table caption.

Response: We thank the reviewer for identifying the unintentional mistake. We tried with different values of the considered parameters. The best values found for each parameter are shown in Table 3. The caption of the table is changed as suggested.

7. Which dataset is used for hyperparameters tuning? 60% of the train set? What cross validation is used for this tuning?

Response: The hyperparameter tuning was done on the training set (i.e., 60%). We applied k-fold (k=6) cross-validation during the training phase itself. A separate subsection (5.4) is added in the manuscript to cover the validation.

8. What method is used for hyperparameters tuning? Grid search method for all algorithms? Kindly include the explanation of the grid search method.

Response: Besides the grid search method, we also tried with random search method but found that the grid search method provided the best results. Also, we preferred the grid search method because it is used in most of the literature for hyperparameter tuning in disease prediction. The further justification for using the grid search method is discussed in Section 5.3.

9. The accuracy results should have both for the train and test set. Figure 8 is a result for which set? The obtained confusion matrix in Figure 7 is for the test set?

Response: We thank the reviewer for the suggestion. We have added the training accuracy along with the testing accuracy in Fig. 8 (now Fig. 9). The confusion matrix shown in Fig. 7 (now Fig. 8) is of only the test set. Adding a confusion matrix of the training set would occupy another whole page. To keep the manuscript length standard, we decided not to add in the main text. If this reviewer still prefers this, we can do next time.

10. Kindly explain what happened in this stage c) Detect and replace the outliers. Why did you replace the outlier? The outlier is your real data.

Response: We thank the reviewer for this valuable point. Outliers are data points that considerably differ from the majority of the data in the dataset. They are generally resulted due to various errors. Outliers can significantly affect how well machine learning models function. They can skew the data distribution, making it non-normal or non-uniform. Therefore, it is important to deal with them. Removing or replacing outliers can make prediction models more robust and less susceptible to extreme values, allowing them to capture the underlying patterns more accurately.

---

## [Decision Letter · Decision Letter 3]

20 Nov 2023

Chronic Kidney Disease Prediction Using Boosting Techniques based on Clinical Parameters

PONE-D-23-02469R3

Dear Dr. Ganie,

We’re pleased to inform you that your manuscript has been judged scientifically suitable for publication and will be formally accepted for publication once it meets all outstanding technical requirements.

Kind regards,

Anwar P.P. Abdul Majeed

Academic Editor

PLOS ONE

Additional Editor Comments (optional):

Reviewers' comments:

Reviewer's Responses to Questions

**Comments to the Author**

1. If the authors have adequately addressed your comments raised in a previous round of review and you feel that this manuscript is now acceptable for publication, you may indicate that here to bypass the “Comments to the Author” section, enter your conflict of interest statement in the “Confidential to Editor” section, and submit your "Accept" recommendation.

Reviewer #4: All comments have been addressed

2. Is the manuscript technically sound, and do the data support the conclusions?

Reviewer #4: Yes

3. Has the statistical analysis been performed appropriately and rigorously? 

Reviewer #4: Yes

4. Have the authors made all data underlying the findings in their manuscript fully available?

Reviewer #4: Yes

5. Is the manuscript presented in an intelligible fashion and written in standard English?

Reviewer #4: Yes

6. Review Comments to the Author

Reviewer #4: All of the comments (10 comments) have been well addressed by the authors. Overall, I am satisfied with the revised article. The article is ready to be published.

7. PLOS authors have the option to publish the peer review history of their article (what does this mean?). If published, this will include your full peer review and any attached files.

Reviewer #4: No

---

## [Editor Report · Acceptance letter]

23 Nov 2023

PONE-D-23-02469R3 

Chronic Kidney Disease Prediction Using Boosting Techniques based on Clinical Parameters 

Dear Dr. Ganie:

I'm pleased to inform you that your manuscript has been deemed suitable for publication in PLOS ONE. Congratulations! Your manuscript is now with our production department. 

Kind regards, 

on behalf of

Dr. Anwar P.P. Abdul Majeed 

Academic Editor

PLOS ONE